# A Single 2D Pose with Context is Worth Hundreds for 3D Human Pose Estimation

**Qitao Zhao**[1]*   **Ce Zheng**[2]   **Mengyuan Liu**[3]   **Chen Chen**[2]

[1]Robotics Institute, Carnegie Mellon University
[2]Center for Research in Computer Vision, University of Central Florida
[3]Key Laboratory of Machine Perception, Peking University, Shenzhen Graduate School

## Abstract

The dominant paradigm in 3D human pose estimation that lifts a 2D pose sequence to 3D heavily relies on long-term temporal clues (*i.e.*, using a daunting number of video frames) for improved accuracy, which incurs performance saturation, intractable computation and the non-causal problem. This can be attributed to their inherent inability to perceive spatial context as plain 2D joint coordinates carry no visual cues. To address this issue, we propose a straightforward yet powerful solution: leveraging the *readily available* intermediate visual representations produced by off-the-shelf (pre-trained) 2D pose detectors – no finetuning on the 3D task is even needed. The key observation is that, while the pose detector learns to localize 2D joints, such representations (*e.g.*, feature maps) implicitly encode the joint-centric spatial context thanks to the regional operations in backbone networks. We design a simple baseline named **Context-Aware PoseFormer** to showcase its effectiveness. *Without access to any temporal information*, the proposed method significantly outperforms its context-agnostic counterpart, PoseFormer [74], and other state-of-the-art methods using up to *hundreds of* video frames regarding both speed and precision. *Project page:* **qitaozhao.github.io/ContextAware-PoseFormer**

## 1   Introduction

3D human pose estimation (HPE) aims to localize human joints in 3D, which has a wide range of applications, including motion prediction [37], action recognition [9], and tracking [47]. Recently, with the large availability of 2D human pose detectors [8, 52], lifting 2D pose sequences to 3D (referred to as lifting-based methods) has been the *de facto* paradigm in the literature. Compared to raw RGB images, 2D human poses (as an intermediate representation) have two essential advantages. On the one hand, domain gaps exist between images (input) and 3D joint locations (output), whereas this is not the case for 2D and 3D joints. Primarily, 2D joint coordinates provide highly task-relevant positional information for localizing joints in 3D. On the other, 2D coordinate representation is exceptionally lightweight in terms of memory cost. For example, in a typical COCO [31] setting, a 2D human pose requires only $17 \times 2$ floating points, whereas an RGB image requires $256 \times 192 \times 3$ (or even more with higher resolution). This property enables state-of-the-art lifting-based methods to leverage extremely long-term temporal clues for advanced accuracy, *e.g.*, 243 video frames for VideoPose3D [45], MixSTE [67], and P-STMO [50]; large as 351 frames for MHFormer [29].

Scaling up the input has brought consistent improvements so far, yet some concerns naturally arise. First, the performance gains seem to level off when excessively long sequences are used as input (*e.g.*, improvements are marginal when the frame number increases from 243 to 351 [29]). Second, long input sequences bring non-trivial computational costs. For instance, temporal processing using transformers (a popular choice) is expensive, especially with large frame numbers. Third, the use of

---

*Work was done while Qitao was an intern mentored by Chen Chen.

37th Conference on Neural Information Processing Systems (NeurIPS 2023).

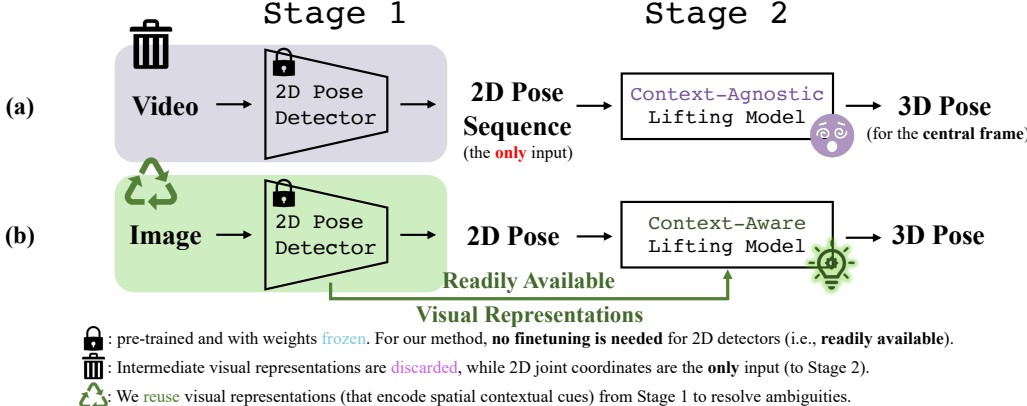

Figure 1: A comparison of existing lifting-based methods with ours at a framework level. (a) Existing methods (video-based): *time-intensive* and *context-agnostic*. (b) Ours (image-based): *time-free* and *context-aware*. We leverage intermediate visual representations from Stage 1. Notably, we do not finetune the 2D detectors for the lifting task, thus easing training and bringing no extra costs.

future video frames renders real-world online applications impossible. Concretely, existing works adopt a 2D pose sequence to estimate the 3D human pose for the *central frame* where half of the frames in the sequence are *inaccessible* for the current time step, which is known as the non-causal problem [45]. *While gains from long input sequences come at a cost, is there any alternative approach we can take to easily boost performance?* We attempt to answer this question in our work.

We start by revisiting the whole pipeline of lifting-based methods, which involves two stages as shown in Fig. 1 (a). In Stage 1, a 2D pose detector estimates human joints for each image from a video clip, with a set of intermediate representations as byproducts, *e.g.*, feature maps of varying resolutions. In Stage 2, the detected 2D pose sequence (output of Stage 1) is lifted to 3D, while such representations are *discarded*. This can be problematic: the (multi-scale) joint-centric spatial context encoded by these feature maps is lost. We argue: *the omission of spatial contextual information is the primary factor contributing to the time-intensive nature of existing lifting-based methods.*

*So, what makes spatial context important?* As an inverse problem, 3D pose estimation inherently suffers from ambiguities [29, 36, 57] such as depth ambiguity and self-occlusion, yet they can be mitigated by utilizing spatial context from the monocular image. For depth ambiguity, the shading difference is an indicator of depth disparity [57] whereas the occluded body parts can be inferred from visible ones with human skeleton constraints [36, 57]. Psychological studies [3, 36] provide evidence that appropriate context helps reduce ambiguities and promote visual recognition for the human perception system. Since 2D keypoints alone are unable to encode such *spatial* context, existing lifting-based approaches, in turn, resort to long-term *temporal* clues to alleviate ambiguities.

We offer a surprisingly straightforward solution to "fix" the established lifting framework as depicted in Fig. 1 (b): retrieving the lost intermediate visual representations learned by 2D pose detectors and engaging them in the lifting process. As previous works limit their research scope to the design of lifting models (Stage 2), the cooperation of both stages is largely under-explored. This approach yields two major benefits: First, such representations encode joint-centric spatial context that promotes reducing ambiguity – the core difficulty in 3D human pose estimation. Second, we illustrate in this work that these representations can be used out of the box, *i.e.*, no finetuning on the 3D task or extra training techniques are needed.

To showcase the effectiveness of our solution, we design a simple transformer-based baseline named **Context-Aware PoseFormer**. The proposed method leverages multi-resolution feature maps produced by 2D detectors in a sparse manner. Specifically, we extract informative contextual features from these feature maps using deformable operations [12, 75] where the detected 2D joints serve as reference points. This helps mitigate the noise brought by imperfect pose detectors while avoiding heavy computation. Furthermore, we design a *pose-context interaction module* to exchange information between the extracted multi-level contextual features and the 2D joint embedding that encodes positional clues. Finally, a *spatial transformer module* is adopted to model spatial dependencies between human joints, which follows PoseFormer [74]. Our approach shows encouragingly strong

performance (see Fig. 2): our single-frame model outperforms 351-frame MHFormer [29] and other state-of-the-art lifting-based methods, highlighting the potential of leveraging contextual information in improving pose estimation accuracy.

Our contributions are summarized as follows:
1) We propose a novel framework that addresses the time-intensive issue present in existing methods by incorporating context awareness. This is achieved by leveraging readily available visual representations learned by 2D pose detectors.
2) We introduce a simple baseline that efficiently extracts informative context features from learned representations and subsequently fuses them with 2D joint embedding that provides positional clues.
3) On two benchmarks (Human3.6M [21] and MPI-INF-3DHP [39]), our single-frame model demonstrates significant performance improvements over both non-temporal and temporal methods that use up to hundreds of video frames.
4) Our work opens up opportunities for more skeleton-based methods to utilize visual representations learned by well-trained backbones from upstream tasks in an out-of-the-box manner.

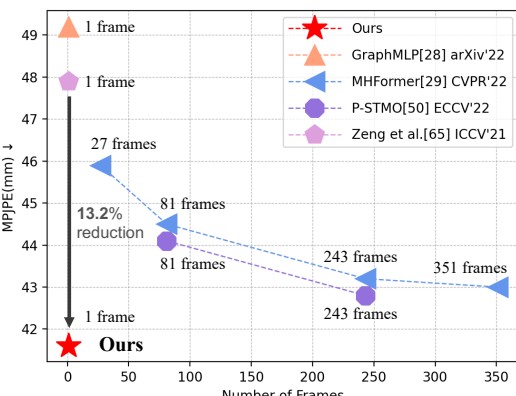

Figure 2: Our single-frame method outperforms both non-temporal and temporal methods that use up to 351 frames on Human3.6M.

## 2  Related Work

Early works [54, 17, 44, 53] estimate the 3D human pose from monocular images without explicitly using the corresponding 2D pose as an intermediate representation. With the rapid progress in 2D pose estimation [42, 8, 52], lifting 2D poses to 3D has been dominant in the literature. As our work aims at improving the lifting-based framework, we primarily introduce works of this line. We refer readers to the recent HPE survey [73] for a more comprehensive background and detailed information.

**Single-frame methods.** Before the surge of deep learning, Jiang [22] performs 3D pose estimation based on nearest neighbors with a large 3D human pose library, and this approach is recently revisited by Chen *et al.* [5] with a closed-form algorithm to project 3D pose exemplars to 2D. Martinez *et al.* [38] design a simple linear-layer model to illustrate that the difficulty of 3D human pose estimation mainly lies in precise 2D joint localization. Moreno-Noguer [41] formulates the task as 2D-to-3D distance matrix regression. To mitigate the demand for 3D annotations, Pavlakos *et al.* [43] leverage ordinal depths of human joints as an extra training signal. Given the human skeleton topology, recent works (*e.g.*, GraphSH [60], HCSF [65], GraFormer [70], GraphMLP [28]) use graph neural networks to reason spatial constraints of human joints, and lift them to 3D.

**Multi-frame methods.** As initial attempts, some works explore temporal cues for robust pose estimation in crowded scenes [1] or temporally consistent results [40]. More recently, a number of works model spatial-temporal dependencies for 2D pose sequences, *e.g.*, using LSTM [24, 59], CNN [45, 6] and GNN [56, 65]. They have achieved superior performance than non-temporal methods. Driven by the great success of vision transformers in image classification [13, 33], object detection [4, 75] and *etc*, transformer-based methods [74, 29, 67, 50, 69, 72] are currently the center of research interest. PoseFormer [74] is the first transformer-based model in the community, which builds up inter-joint correlations within each video frame and human dynamics across frames with transformer encoders, respectively. Due to its straightforward architecture, PoseFormer is highly malleable and rapidly gains a series of follow-ups [67, 69]. Our work presents a simple *single-frame* baseline model where the *inter-joint modeling module* is adapted from the spatial encoder of PoseFormer.

**Fusing image features with positional joint clues.** Compared to previous lifting-based works, our approach actively engages visual representations from 2D pose detectors in the lifting process, thus being *context-aware* and achieving promising results. Notably, the fusion of image features with positional information about human joints has been explored in some works but within differing contexts or with distinct motivations. Tekin *et al.* [55] propose to fuse image features with 2D joint heatmaps, where 2D joints are not explicitly regressed as in the lifting-based pipeline. Closer to

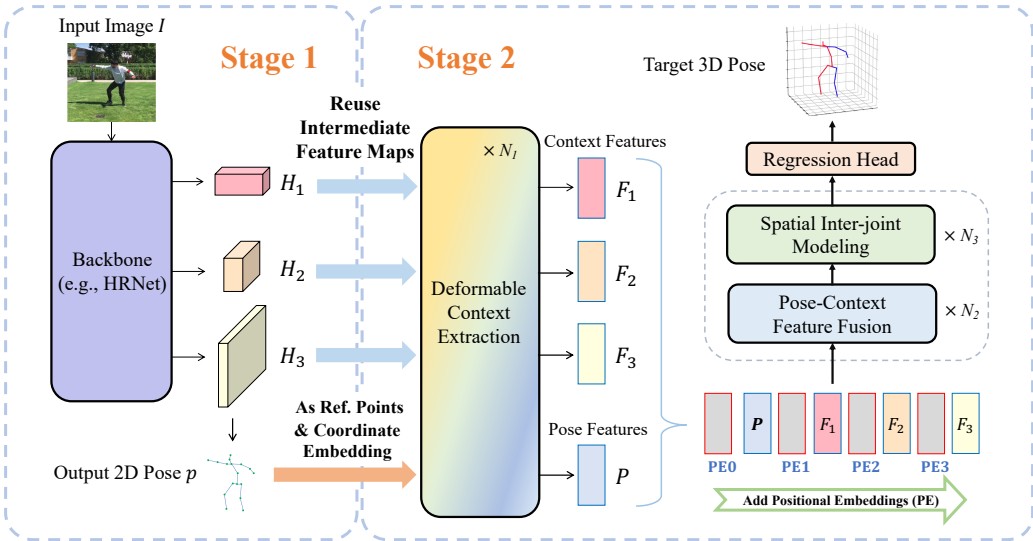

Figure 3: An overview of Context-Aware PoseFormer. Stage 1 (left): The 2D pose detector estimates the 2D pose, with a set of feature maps as byproducts. In Stage 2 (right), we extract informative join-context features from such feature maps and subsequently fuse them with 2D pose features.

our work, Zhao *et al.* [68] extend their graph network (SemGCN) by incorporating image features. Specifically, image features are pooled at the detected joints and then concatenated together with joint coordinates, serving as the input to SemGCN. Though this approach is similar to ours at first glance, our work differs in that 1) our method needs no multi-stage training, 2) we consider the uncertainty associated with 2D joints in joint-context extraction, 3) we have designed dedicated modules to handle the domain gap between image features and joint coordinates, and 4) our performance surpasses that of SemGCN by a significant margin. In addition to technical details and performance comparisons, our underlying motivation for this research also differs. More discussion about relevant literature can be found in Appendix A.

## 3 Method

We propose a new framework to solve the time-intensive problem of previous lifting-based methods. This is achieved by incorporating the spatial context (encoded by intermediate feature maps from 2D pose detectors) into the 2D-to-3D lifting process (see also Fig. 1 (b)). We design a simple baseline model named **Context-Aware PoseFormer** to exemplify the gains brought by this approach (an overview is in Fig. 3). In the following, we first briefly introduce the context-agnostic counterpart of our method, PoseFormer [74]. Then, we provide detailed descriptions of our approach.

### 3.1 A Context-agnostic Counterpart: PoseFormer

PoseFormer [74] is one of the first transformer-based models for 3D human pose estimation, which mainly consists of two modules: the spatial encoder and the temporal encoder. PoseFormer processes an input 2D pose sequence as follows. First, the sequence $s \in \mathbb{R}^{F \times J \times 2}$ is projected to a latent space indicated by $S \in \mathbb{R}^{F \times J \times C}$, where $F$, $J$ and $C$ denote the sequence length, joint number, and embedding dimension respectively. Learnable (zero-initialized) spatial positional embedding $E_{spa} \in \mathbb{R}^{1 \times J \times C}$ is summed with $S$ to encode joint-specific information. Second, the spatial encoder models inter-joint dependencies at each video frame (time step) with cascaded transformers. Third, the output of the spatial encoder $Z \in \mathbb{R}^{F \times (J \times C)}$ is flattened as the input to the temporal encoder. Similarly, temporal positional embedding $E_{temp} \in \mathbb{R}^{F \times (J \times C)}$ is added to $Z$ to encode the information related to frame indices. The temporal encoder builds up frame-level correlations with transformer layers in this stage. Finally, a 1D convolution layer is adopted to gather temporal information, and a linear layer outputs the target 3D pose $y \in \mathbb{R}^{1 \times (J \times 3)}$ for the central video frame. PoseFormer and its follow-ups [67, 50, 69] are inherently limited to deal with ambiguities since they barely receive 2D joint coordinates as input. Due to the simplicity, scalability, and effectiveness of PoseFormer, we choose PoseFormer as the

basis of our method, boosting it to be context-aware and free of temporal information. Detailed descriptions of our method are presented below.

## 3.2 Context-Aware PoseFormer

Our core insight is retrieving the spatial context encoded by readily available feature maps from 2D human pose detectors. As the multi-stage design promotes 2D human pose estimation [52, 11, 63], our method also benefits from multi-resolution feature maps. An illustration of model input is on the left of Fig. 3. For a given RGB image $I$ of size $H \times W \times 3$, an off-the-shelf 2D pose detector produces the corresponding 2D human pose $p \in \mathbb{R}^{J \times 2}$, and a set of intermediate feature maps with varying resolutions as byproducts, $\{H_l \in \mathbb{R}^{H_l \times W_l \times C_l}\}_{l=1}^{L}$ (where $L$ is the total number of feature maps). Since stacked network layers and down-sampling operations increase the receptive field (and correspondingly the scale of context) of each unit on feature maps [35], high-resolution feature maps encode fine-grained visual cues (*e.g.*, joint existence) while low-resolution ones tend to keep high-level semantics (*e.g.*, human skeleton structures). During the process that 2D pose detectors are trained to localize human joints, multi-level feature maps implicitly encode information about the spatial configurations of joints.

**How to effectively utilize these feature maps** is a non-trivial question. Blindly processing them in a global way using, *e.g.*, convolutions [20], or vision transformers [13], may bring unnecessary computational overhead since the background that takes up a large proportion of the image does not provide task-relevant features related to human joints. The most straightforward approach to treat feature maps in a preferably sparse manner is to sample feature vectors at the detected joints. However, in real scenarios, 2D pose detectors inevitably introduce errors. Therefore, the context feature obtained by such an approach may not well represent the spatial context associated with the joint of our interest, leading to suboptimal performance. We propose a *Deformable Context Extraction* module to deal with this issue, which not only attends to the detected joints but also the regions around them. As the detected 2D joints are not the only source of contextual features, the errors brought by 2D pose detectors can be mitigated. Moreover, we design a *Pose-Context Feature Fusion* module to fuse the features of two modalities: spatial contextual features and 2D joint embedding that encodes positional clues about human joints. Finally, *Spatial Inter-joint Modeling* is adopted to build up dependencies across human parts. They are organized hierarchically (on the right, Fig. 3). The effectiveness of each element is verified in Sec. 4.3.

**Deformable Context Extraction.** This module uses deformable attention [75] to extract informative spatial contextual cues from feature maps. Specifically, for each detected joint, we produce a set of sampling points on multi-scale feature maps whose offsets and weights are learned based on the features of reference points (*i.e.*, the detected joint of interest), denoted by $\{F_l\}_{l=1}^{L}$. We make learned offsets and weights "reference-position-aware" by adding the embedding of 2D joint coordinates $P$ to source features (after they are projected to a shared dimension $C$). Let $l$ index a feature level and $j$ index a human joint, and Deformable Context Extraction is formulated as:

$$
\begin{aligned}
F_{lj}^{'n} &= \text{DeformAttn}(F_{lj}^{n-1} + P_j) + F_{lj}^{n-1}, & F_{lj}^{n-1}, P_j \in \mathbb{R}^C, \ n = 1 \dots N_1 \\
F_{lj}^{n} &= \text{MLP}(F_{lj}^{'n}) + F_{lj}^{'n}, & F_{lj}^{n}, F_{lj}^{'n} \in \mathbb{R}^C, \ n = 1 \dots N_1 \\
\text{DeformAttn}(F_{lj}^{n-1} + P_j) &= \sum_{m=1}^{M} \Big[ \sum_{k=1}^{K} A_{lmk} \cdot W_{lm} H_l(p_j + \Delta p_{lmk}) \Big], & H_l(p_j + \Delta p_{lmk}) \in \mathbb{R}^{C_l}
\end{aligned}
\tag{1}
$$

where $m$ iterates over the attention heads, $k$ over the sampled points around the detected joint $p_j$, and $K$ is the total sampling point number. For the $l^{\text{th}}$ feature map, $\Delta p_{lmk}$ represents the sampling offset of the $k^{\text{th}}$ sampling point in the $m^{\text{th}}$ attention head, while $A_{lmk}$ denotes its corresponding attention weight. We provide visualization of sampling points learned by deformable attention in Sec. 4.3, where the learned sampling points attempt to discover ground truth 2D joints despite the noise brought by pose detectors. More details and visualization about this module are available in Appendix E.

**Pose-Context Feature Fusion.** This module aims at exchanging information between two modalities (*i.e.*, pose embedding and the extracted spatial context) and between different levels of context features at the same time. Cross-modality learning gives rise to powerful vision-language models [23, 2], while the multi-scale design has proved beneficial to a large set of computer vision tasks

[15, 16, 71]. Inspired by multi-modality models [2] that feed visual tokens and language tokens into a unified transformer encoder, we naturally adapt this design to model interactions between pose embedding and multi-level context features, where transformers learn a joint representation for both modalities [51, 7]. The implementation of this module is:

$$
\begin{aligned}
\boldsymbol{X}_j^0 &= \text{Stack}([\boldsymbol{P}_j, \boldsymbol{F}_{1j}^{N_1}, \ldots, \boldsymbol{F}_{Lj}^{N_1}], \ dim = 0), & \boldsymbol{P}_j \in \mathbb{R}^C, \ \boldsymbol{F}_{1j}^{N_1}, \ldots, \boldsymbol{F}_{Lj}^{N_1} \in \mathbb{R}^C \\
\boldsymbol{X}_j^{'n} &= \text{SelfAttn}(\boldsymbol{X}_j^{n-1}) + \boldsymbol{X}_j^{n-1}, & \boldsymbol{X}_j^{n-1} \in \mathbb{R}^{(L+1) \times C}, \ n = 1 \ldots N_2 \quad (2) \\
\boldsymbol{X}_j^n &= \text{MLP}(\boldsymbol{X}_j^{'n}) + \boldsymbol{X}_j^{'n}, & \boldsymbol{X}_j^{'n} \in \mathbb{R}^{(L+1) \times C}, \ n = 1 \ldots N_2
\end{aligned}
$$

where $j$ $(1 \ldots J)$ indicates that feature fusion is performed for each joint. Transformer layers reduce domain gaps for both modalities in shared latent space [25] and promote message-passing where joint-position information and multi-level contextual cues complement each other. In addition, we introduce Unified Positional Embedding $\boldsymbol{E}_{uni} \in \mathbb{R}^{(L+1) \times J \times C}$ ($L$ for joint-context feature vectors and 1 for 2D pose embedding) to encode modality-related and joint-specific information simultaneously.

**Spatial Inter-joint Modeling.** With the two modules above, elegant local representations are learned for each joint individually. To understand the human skeleton system and its corresponding spatial context with a global view, inter-joint dependencies are modeled based on the learned per-joint features using a spatial transformer encoder following PoseFormer [74]. The spatial encoder in PoseFormer simply receives the linear projection of (context-agnostic) 2D joint coordinates as input, whereas our joint-level representations are enhanced with spatial contextual cues (thus being context-aware). The output of Pose-Context Feature Fusion module $\{\boldsymbol{X}_j^{N_2}\}_{j=1}^J$ is flattened and stacked as input to the $N_3$-layer transformer encoder. Each joint token ($J$ in total) is of $(L+1) \times C$ dimensions, encoding both positional and contextual information for the related joint.

**Output and loss function.** A simple linear layer is adopted to obtain the final 3D pose $\boldsymbol{y} \in \mathbb{R}^{J \times 3}$. We use $L_2$ loss as PoseFormer [74] to supervise the estimated result.

**Discussion.** Our model follows a local-to-global hierarchy. Specifically, local representations are first learned for each joint, on which inter-joint modeling is subsequently performed. This design provides our model with preferable interpretability and scalability: modules are inherently disentangled according to the dependencies they attempt to learn and can be easily extended or replaced. Plus, the local-to-global organization avoids the heavy computation brought by globally modeling all elements at once. These virtues make our method a desirable stepping stone for future research.

## 4    Experiments

Our method, Context-Aware PoseFormer, is referred to as "CA-PF-`backbone name`" in tables. We conduct experiments on two benchmarks (Human3.6M [21] and MPI-INF-3DHP [39]). Human3.6M [21] is the most popular benchmark for 3D human pose estimation, where more than 3.6 million video frames are captured indoors from 4 different camera views. MPI-INF-3DHP [39] includes videos collected from indoor scenes and challenging outdoor environments. Both datasets provide subjects performing various actions with multiple cameras. On Human3.6M, we report MPJPE (Mean Per Joint Position Error, where the Euclidean distance is computed between estimation and ground truth) and PA-MPJPE (the aligned MPJPE). For MPI-INF-3DHP, we report MPJPE, Percentage of Correct Keypoint (PCK) within the 150mm range, and Area Under Curve (AUC). Settings follow previous works [74, 29, 67, 50]. Due to page limitations, we place details about our implementations in Appendix B.

### 4.1    Comparison on Human3.6M

**Accuracy**. We compare our single-frame method with both single-frame and multi-frame methods on Human3.6M [21] (Table 1). Previous methods receive CPN-detected [8] 2D joints as input. We also follow this setting to ensure a fair comparison. Benefiting from spatial contextual clues encoded by the CPN pose detector, our approach outperforms state-of-the-art single-frame methods by a huge margin. For instance, Zeng *et al.* [65] obtain 47.9mm MPJPE, whereas our CA-PF-CPN achieves 41.6mm (a **13.2%** error reduction). This result has already surpassed the performance of most modern multi-frame methods. To show the *flexibility* of our method and to further investigate spatial context

Table 1: Comparison with both single-frame (top) and multi-frame (middle) methods on Human3.6M. MPJPE is reported in millimeters. The best results are in bold, and the second-best ones are underlined.

| | Method | Venue | 2D Pose Detector | Frame | FLOPs (**G**) for the Lifting Module | MPJPE | PA-MPJPE |
|---|---|---|---|---|---|---|---|
| Single Frame | GraphSH [60] | CVPR'21 | CPN [8] | 1 | - | 51.9 | - |
| | HCSF *et al.* [65] | ICCV'21 | CPN [8] | 1 | - | 47.9 | 39.0 |
| | GraFormer [70] | CVPR'22 | CPN [8] | 1 | - | 51.8 | - |
| | GraphMLP [28] | arXiv'22 | CPN [8] | 1 | 0.3 | 49.2 | 38.6 |
| Multi-frame | Pavllo *et al.* [45] | CVPR'19 | CPN [8] | 243 | 0.03 | 46.8 | 36.5 |
| | Liu *et al.* [32] | CVPR'20 | CPN [8] | 243 | - | 45.1 | 35.6 |
| | Wang *et al.* [56] | ECCV'20 | CPN [8] | 96 | - | 45.6 | 35.5 |
| | Zeng *et al.* [64] | ECCV'20 | CPN [8] | 243 | - | 44.8 | 34.9 |
| | PoseFormer [74] | ICCV'21 | CPN [8] | 81 | 1.4 | 44.3 | 34.6 |
| | StridedTrans. [27] | TMM'22 | CPN [8] | 351 | 2.1 | 43.7 | 35.2 |
| | MHFormer [29] | CVPR'22 | CPN [8] | 351 | 14.2 | 43.0 | 34.4 |
| | MixSTE [67] | CVPR'22 | CPN [8] | 243 | 277.4 | 40.9 | **32.6** |
| | P-STMO [50] | ECCV'22 | CPN [8] | 243 | 1.5 | 43.0 | 34.4 |
| | Einfalt *et al.* [14] | WACV'23 | CPN [8] | 18 | 1.0 | 45.0 | 36.3 |
| | CA-PF-CPN (ours) | | CPN [8] | 1 | 0.6 | 41.6 | 33.9 |
| | CA-PF-HRNet-32 (ours) | | HRNet-32 [52] | 1 | 0.6 | 41.4 | 33.5 |
| | CA-PF-HRNet-48 (ours) | | HRNet-48 [52] | 1 | 0.6 | **39.8** | 32.7 |

from various backbones, we change the backbones that generate feature maps: our CA-PF-HRNet-32 [52] achieves 41.4mm MPJPE; with a larger backbone HRNet-48, the error reduces to 39.8mm, which is superior to *all* previous multi-frame methods. For example, this result outperforms MixSTE [67] with 243 frames (39.8 v.s. 40.9mm, a 2.7% error reduction) and MHFormer [29] with even 351 frames (39.8 v.s. 43.0mm, a **7.4**% error reduction). Our method presents highly competitive accuracy in the absence of dense temporal modeling. Moreover, our method consistently gains improvements from advanced 2D pose detectors. We expect our approach's performance to increase further with powerful large visual foundation models, as such foundation models have been attracting increasing attention and interest.

**Computational cost**. Our method requires far less computation than state-of-the-art methods (see also Table 1). For example, with similar accuracy, MHFormer [29] takes 14.2 GFLOPs, whereas our CA-PF-CPN [8] demands only 0.6 GFLOPs (about **24**× reduction). Two reasons account for the lightweight property of the proposed approach: (1) Non-temporal input removes large computation for dense temporal modeling. Modeling temporal dependencies using *e.g.*, transformers [74, 67] is computationally heavy especially when the frame number is increased for improved accuracy. (2) We utilize multi-scale feature maps (from pose detectors) in a sparse manner (*i.e.*, with deformable attention [75]). We avoid applying global operations on the whole feature maps, which may introduce unnecessary computational budgets on uninformative backgrounds.

## 4.2 Comparison on MPI-INF-3DHP

In Table 2, the comparison is also conducted with both single-frame (top) and multi-frame (middle) methods on the MPI-INF-3DHP dataset [39]. We use ground truth 2D pose detection as input, following [74, 29, 50], and HRNet-32, HRNet-48 [52] are adopted as the backbone network to generate multi-resolution feature maps. Our approach outperforms other methods, including the state-of-the-art multi-frame one, P-STMO [50]. P-STMO uses 81 video frames as input, with additional masked joint self-supervised pre-training. In contrast, our approach has no access to temporal information and does not need an extra pre-training stage. The results verify the generalization ability of our method to different datasets, particularly in challenging outdoor environments.

Table 2: Comparison on MPI-INF-3DHP. *T* indicates the number of video frames used by models.

| Method | PCK ↑ | AUC ↑ | MPJPE ↓ |
|---|---|---|---|
| Xu *et al.* [60] (*T*=1) | 80.1 | 45.8 | - |
| Li *et al.* [26] (*T*=1) | 81.2 | 46.1 | 99.7 |
| Zeng *et al.* [65] (*T*=1) | 82.1 | 46.2 | - |
| PoseFormer [74] (*T*=9) | 95.4 | 63.2 | 57.7 |
| MHFormer [29] (*T*=9) | 93.8 | 63.3 | 58.0 |
| MixSTE [67] (*T*=27) | 94.4 | 66.5 | 54.9 |
| P-STMO [50] (*T*=81) | 97.9 | 75.8 | 32.2 |
| CA-PF-HRNet-32 (ours) | 98.0 | 75.4 | 32.7 |
| CA-PF-HRNet-48 (ours) | **98.2** | **76.3** | **31.4** |

Table 3: Ablation study on spatial context and each component of our method. Experiments are conducted on Human3.6M with HRNet-32 as the backbone. MPJPE is reported in millimeters.

| Step | Context-aware | Inter-joint Modeling | Pose-Context Feature Fusion | Deform. Cont. Extraction | FLOPs (**M**) | MPJPE ↓ |
|------|---------------|----------------------|------------------------------|--------------------------|---------------|---------|
| (0) | ✗ | ✓ | ✗ | ✗ | 446.0 | 51.2 |
| (1) | ✓ | ✓ | ✗ | ✗ | 448.0 | 43.5 (7.7↓) |
| (2) | ✓ | ✓ | ✓ | ✗ | 537.3 | 42.8 (8.4↓) |
| (3) | ✓ | ✓ | ✓ | ✓ | 609.9 | 41.4 (9.8↓) |

## 4.3 Ablation Study

**Gains from spatial context and ablations on model components.** In this section, we first investigate the gains *purely* brought by spatial contextual clues from 2D pose detectors and then the effectiveness of each component in our method (please refer to Sec. 3.2 for more details). Specifically, we show how the context-agnostic counterpart of our method, (single-frame) PoseFormer [74] is converted to Context-Aware PoseFormer step by step (step-wise improvements are shown in Table 3).

- **Step 0, Context-Agnostic:** For PoseFormer, 2D joint coordinates detected by a pose detector are linearly embedded to dimension $C$ as per-joint representation (*i.e.*, joint embedding), and Inter-joint Modeling is subsequently performed to build up correlations across the joint embeddings. Plain 2D coordinates encode no spatial context for joints that helps reduce ambiguity, PoseFormer (and other lifting-based methods) is thus referred to as "context-agnostic". This serves as the starting point of our method.
- **Step 1, Context-Aware but simple concatenation:** The first step to make PoseFormer "context-aware" is to incorporate joint-context features into its per-joint representation. We retrieve the multi-scale feature maps produced by the 2D pose detector, simply sample feature vectors on the detected joint locations from each feature map, and project them to dimension $C$. For each joint, the sampled context feature vectors that encode joint-centric visual cues are concatenated together with the coordinate embedding that encodes positional information. Even naive involvement (as simple as concatenation) of spatial context clues brings a huge improvement as shown in Table 3: a **15.0**% error reduction (from 51.2 to 43.5mm), which demonstrates that *leveraging the readily available visual representations learned by upstream backbones is effective and promising*.
- **Step 2, Context-Aware with Pose-Context Feature Fusion:** We promote cross-modality feature fusion by applying transformers to joint embeddings and the sampled multi-scale context features before concatenation, which brings another 1.6% error reduction.
- **Step 3, Final version of our Context-Aware PoseFormer:** The context features in step 1 are obtained by directly sampling at the detected joints, which leads to suboptimal performance since 2D pose detectors inevitably introduce noise. Thus, the sampled feature vectors can not well represent the context associated with the joint of interest. We introduce Deformable Context Extraction to mitigate this issue, which adaptively attends to the detected joint along with its surrounding regions using learnable sampling points, which further reduces the error by 3.3%.

We also offer visualization of learned sampling points (brighter colors denote larger attention weights) in Fig. 4, where they try to approach the ground truth despite the noisy detected input. Note that in training we do *not* supervise them using ground truth 2D pose. More visualization is in Appendix E.

**What kind of spatial context are we looking for?** In this part, we go a step further to explore what context is desirable to improve 3D pose estimation accuracy. Our method benefits from multi-resolution feature maps produced by 2D detectors [52, 11, 63], which encode spatial contextual clues of various levels for human joints. For example, HRNet [52] outputs four feature maps in the last stage, which are 4×, 8×, 16×, 32× downsampled compared to the input image size, respectively. We investigate which one of the four is most informative for our task via ablation experiments (in Table 4, MPJPE is reported on Human3.6M). Removing 4× and 16× downsampled feature

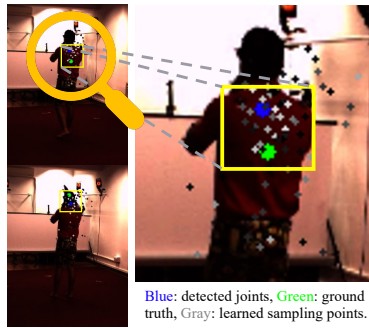

Blue: detected joints, Green: ground truth, Gray: learned sampling points.

Figure 4: In the proposed Deformable Context Extraction module, sampling points attempt to approach the ground truth despite unreliable 2D joint detection.

maps brings the most significant performance drop (a 3.6% and 4.1% error increase for each). In contrast, only a 1.0% error increase is observed when the 32× downsampled feature map is eliminated.

Generally, high-resolution feature maps contribute more to 3D pose estimation than low-resolution ones. The reason is that low-resolution feature maps tend to keep the high-level semantics of joints, such as join type, which may not be relevant to our task – to localize human joints in 3D, where 2D-to-3D joint correspondence is pre-defined. However, we can still gain from low-resolution feature maps as they encode most wide-range spatial dependencies, promoting an understanding of the overall human skeleton structure.

Table 4: Ablation study on feature resolutions.

| 4× | 8× | 16× | 32× | MPJPE ↓ |
|---|---|---|---|---|
| ✓ | ✓ | ✓ | ✓ | 41.4 |
| ✗ | ✓ | ✓ | ✓ | $42.9_{\Delta 1.5}$ |
| ✓ | ✗ | ✓ | ✓ | $42.4_{\Delta 1.0}$ |
| ✓ | ✓ | ✗ | ✓ | $43.1_{\Delta 1.7}$ |
| ✓ | ✓ | ✓ | ✗ | $41.8_{\Delta 0.4}$ |

**What pre-trained features are more desirable?** ImageNet [48] pre-trained backbones (*e.g.*, ResNet [20]) profit a series of downstream tasks [30, 10, 58], yet this seems not applicable to 3D human pose estimation. In Table 5, we replace COCO pre-trained backbones in our method with ImageNet classification pre-trained ones, showing a remarkable performance drop. This should be attributed to the large gap between the pre-training task (image classification) and the downstream task (3D human pose estimation). Exploring more suitable pre-trained features that can transfer their effectiveness into the 3D HPE task is a promising direction. We place more ablations on backbones and pre-training datasets in Appendix D.

Table 5: Ablation study on pre-training tasks.

| Backbone | Pre-training | MPJPE ↓ |
|---|---|---|
| ResNet-50 | 2D Pose | 45.0 |
| | Image Class. | $51.4_{\Delta 6.4}$ |
| HRNet-32 | 2D Pose | 41.4 |
| | Image Class. | $45.8_{\Delta 4.4}$ |
| HRNet-48 | 2D Pose | 39.8 |
| | Image Class. | $43.9_{\Delta 4.1}$ |

## 4.4 Visualization

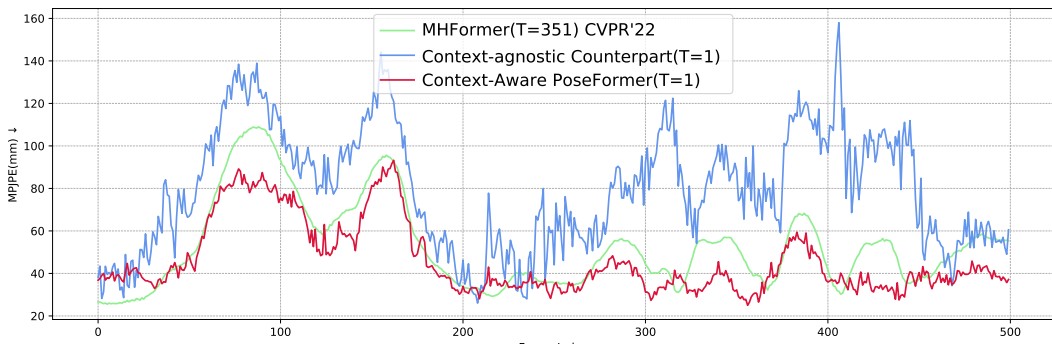

Figure 5: Frame-wise comparison with MHFormer [29] and our context-agnostic counterpart (Sec. 4.3) on Human3.6M test set (S9 Discussion). $T$ indicates the frame number used by models.

**Spatial context promotes temporal stability.** We show a frame-wise comparison on the Human3.6M [21] test set in Fig. 5. Our single-frame approach outperforms competitive MHFormer [29] using 351 frames. Moreover, our context-agnostic counterpart (Sec. 4.3) suffers from temporal instability (*i.e.*, zigzags and error peaks), whereas our method presents a more stable trend. We also provide a qualitative comparison with both methods in Fig. 6. We include more visual comparison on both Human3.6M and MPI-INF-3DHP in Appendix F.

**Spatial context improves the robustness against occlusions**. We provide a comparison with PoseFormer [74] on in-the-wild videos in Fig. 7. We choose two sets of consecutive video frames where the 2D joint detection fails due to strong self-occlusion. Since PoseFormer only accepts 2D joints as input, its estimation is sensitive to the noise of input 2D poses. On the contrary, our method leverages spatial context from images in addition to 2D joint coordinates to localize joints in 3D. Thus, our method shows more reasonable and robust results despite noisy or even outlier 2D joints.

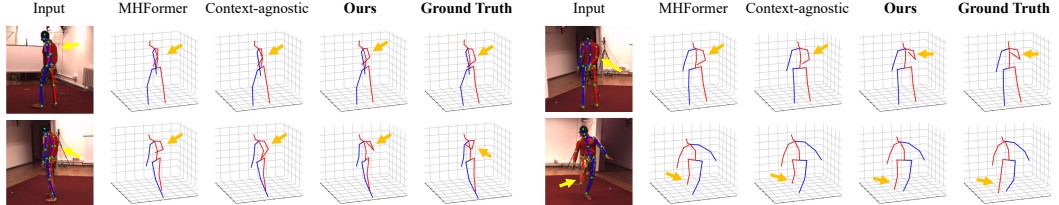

Figure 6: Visual comparison in hard cases, *e.g.*, self-occlusion and noisy 2D detection. We include more qualitative results in Appendix F.

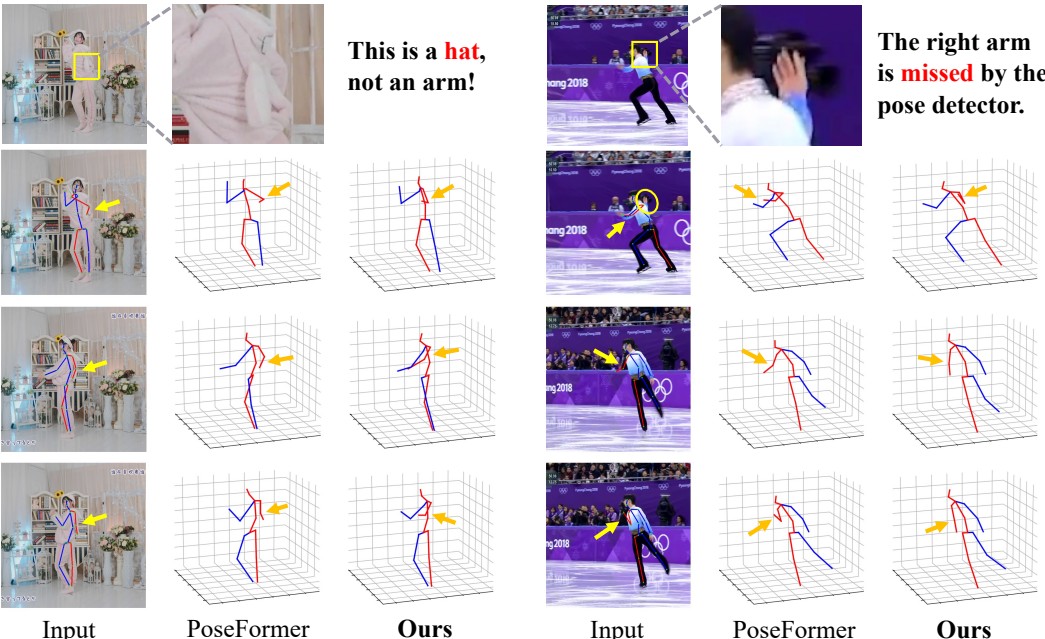

Figure 7: Comparison with PoseFormer [74] on in-the-wild videos. The 2D pose detector fails to localize 2D joints, given *confusing clothing* (left) and severe *self-occlusion* (right). Our method is more robust in such hard cases. False joint detection is indicated by yellow arrows, and the corresponding 3D joint estimation is indicated by orange arrows.

## 5    Conclusion and Discussion

This work presents a new framework that leverages readily available visual representations learned by off-the-shelf 2D pose detectors. Our approach removes the need for extensive long-term temporal modeling excessively employed by existing 3D human pose estimation methods. We further propose a straightforward baseline to showcase its effectiveness: our single-frame model significantly outperforms other single-frame and even multi-frame methods using up to 351 frames while being far more efficient. Our work opens up opportunities for future research to utilize visual representations from well-trained upstream backbones in an out-of-the-box manner.

**Limitations.** We observed in Sec. 4.4 that incorporating the spatial context of joints improves temporal stability, enhancing consistency and smoothness in the estimated results, even without access to explicit temporal clues. However, we acknowledge for all single-frame methods, including ours, mitigating jitters remains a challenge compared to multi-frame methods that leverage temporal clues. This is primarily due to the non-temporal nature of single-frame methods. A potential solution is to extend our method to model *short-term* temporal dependencies, which should not introduce unacceptably high costs. We include some preliminary results in Appendix C. Compared to standard lifting-based methods, the memory cost of our method can be higher as we additionally process image features in the lifting stage. Leveraging the visual context of joints in a more lightweight way is our future research direction.

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

## Appendix

## A    Additional Discussion on Related Work

**Involving 2D joint heatmaps** in image-based 3D human pose estimation has also been explored in some works [62, 18]. Yin *et al.* [62] split the 3D human pose estimation into two sub-tasks: 2D pose estimation and depth regression. Habibie *et al.* [18] disentangle features for the 2D pose from global image features to exploit large-scale 2D pose datasets. These approaches do not explicitly learn features from 2D poses and lift them to 3D.

**Keypoint-based feature extraction.** IVT [46] formulates video-based multi-person 3D pose estimation as an end-to-end framework. The features of each instance in the video are gathered by learning joint offsets from the body center. Compared to IVT, our work follows the standard two-stage pipeline of 3D HPE, and the estimated 2D joints serve as reference points in feature extraction. Pixel-aligned features [49, 66] in the context of 3D human reconstruction use point-wise image features similar to our approach. Their feature extraction is based on the re-projection of estimated 3D meshes [66] or query 3D points [49] instead of 2D human joints from off-the-shelf 2D pose detectors. Pixel-aligned features primarily aim at reducing the misalignment between estimated 3D representations and input 2D images. In contrast, we propose joint-context feature extraction to reduce ambiguities in lifting-based HPE and alleviate current methods' reliance on temporal information.

## B    Implementation Details

**2D pose detector settings.** Our method's overall pipeline includes an off-the-self 2D pose detector and a lifting model. For the first stage, the 2D pose detector is pre-trained on the COCO [31] dataset, without finetuning on the 2D poses from 3D pose estimation datasets, *i.e.*, Human3.6M [21] and MPI-INF-3DHP [39]. We use $256 \times 192$ resolution for input images, and we found higher image resolution improves performance. For 2D-to-3D lifting, the weights of pre-trained 2D detectors are frozen, *i.e.*, no finetuning on the 3D task is needed. This approach makes our method preferably flexible – our method is compatible with a wide range of off-the-shelf (pre-trained) 2D pose detectors. We show in Sec. 4 that our method gains consistent improvements by increasing the capability of 2D pose detectors. In the future, we may leverage more advanced 2D pose detectors to further improve the performance upper bound.

**Lifting model settings.** Our lifting model includes three basic modules: *Deformable Context Extraction*, *Pose-Context Feature Fusion* and *Spatial Inter-joint Modeling*. The layer number of each module is set to 4, following PoseFormer [74]. The hidden dimension (a shared projection dimension $C$) of the model is 128. We use 8 heads in self-attention for transformer layers.

**Training details.** The experiments are conducted on a single NVIDIA RTX 3090 GPU. Our lifting model is trained using the AdamW [34] optimizer for 50 epochs with a batch size of 512. The initial learning rate is 6.4e-3 with an exponential learning rate decay schedule, and the decay factor is 0.99.

## C    A Simple Temporal Extension of Our Method

This section shows that our single-frame method can naturally extend to model temporal dependencies (an overview is in Fig. 8). As illustrated in Sec. 3.2, the *Spatial Inter-joint Modeling* module outputs a feature vector of dimension $(L + 1) \times C$ for each joint, where $L$ is the number of multi-scale feature maps and $C$ is a shared projection dimension of the model. We use a *Temporal Transformer* to model temporal correlations of each joint independently. Specifically, for *Temporal Transformer*, the input token number is the total frame number, and each token is of dimension $(L + 1) \times C$. Using transformers to build up temporal dependencies is straightforward, and this approach has been adopted by PoseFormer [74], MixSTE [67], *etc*. The output of the temporal transformer encoder can be denoted by $\boldsymbol{Z}_{temp} \in \mathbb{R}^{J \times F \times [(L+1) \cdot C]}$, where $J$ is the joint number and $F$ is the frame number. Following PoseFormer, we use 1D convolution to reduce its temporal dimension (gather temporal information) and a linear layer to obtain the final estimated 3D pose $\boldsymbol{y} \in \mathbb{R}^{J \times 3}$.

**Quantitative results**. Due to limited computational resources, we prepare contextual feature vectors (that are sampled at detected joints from multi-level feature maps) in advance and remove the

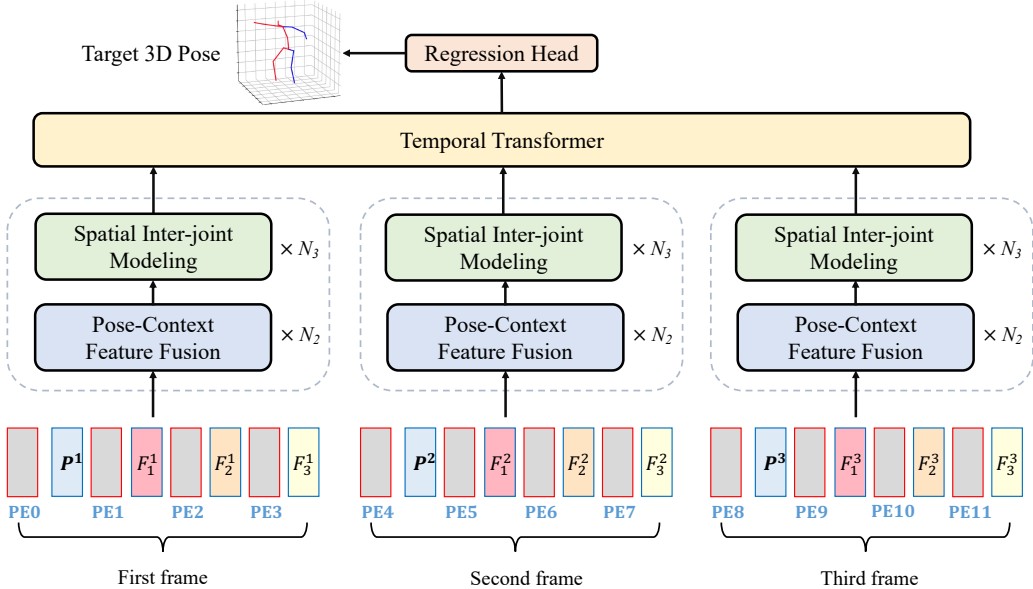

Figure 8: Architecture of our method's simple temporal extension (3 frames as input).

Table 6: We compare the (short-term) temporal extension of our small model (CA-PF-S) with PoseFormer. MPJPE: Mean Per Joint Position Error, the precision metric. MPJVE: Velocity Error, the temporal smoothness metric. The results are reported on Human3.6M (in millimeters).

| Model | Frame | MPJPE ↓ | MPJVE ↓ |
|---|---|---|---|
| PoseFormer | 1 | 53.2 | 13.7 |
| | 3 | 51.0 | 7.1 |
| | 9 | 49.9 | 4.8 |
| | 81 | 44.3 | 3.1 |
| CA-PF-S | 1 | 44.7 | 8.5 |
| | 3 | $44.2_{\downarrow 0.5}$ | $4.8_{\downarrow 3.7}$ |
| | 9 | $43.4_{\downarrow 1.3}$ | $3.4_{\downarrow 5.1}$ |
| | 27 | $40.2_{\downarrow 4.5}$ | $2.1_{\downarrow 6.4}$ |

*Deformable Context Extraction* module. Therefore, the input to our model is discrete feature vectors instead of image sequences. This approach largely reduces the memory cost. We also reduce the shared projection dimension $C$ in our model to speed up training. This small variant of our model is referred to as "CA-PF-S", which has fewer FLOPs compared to our full model "CA-PF" presented in Sec. 4. To show the gains in precision and temporal smoothness from temporal modeling, we report two metrics, MPJPE (Position Error, the precision metric) and MPJVE (Velocity Error, the temporal smoothness metric), on Human3.6M [21]. We compare it with PoseFormer [74] and the results are in Table 6: **First,** increasing the number of input frames brings consistent improvements in both precision and temporal smoothness. For example, by using only 3 video frames, the MPJVE of our method decreases from 8.5 to 4.8mm (a **43.5**% reduction), and the MPJPE is reduced by 1.1%. This indicates that even short-term temporal modeling largely mitigates jitters in estimated results and further improves precision upper bound. **Second,** considering the same number of input frames, our CA-PF-S consistently outperforms PoseFormer in terms of both MPJPE and MPJVE. Moreover, our 3-frame CA-PF-S has already achieved superior MPJPE to 81-frame PoseFormer, and our 9-frame CA-PF-S achieves comparable MPJVE with 81-frame PoseFormer. The results verify the two benefits of spatial contextual clues from 2D pose detectors – accuracy and temporal stability.

Based on the results above, we expect that our method can be successfully extended to model even longer-term temporal dependencies (*e.g.*, 81 video frames) to further boost precision and temporal smoothness. However, modeling such long input sequences with acceptable computational costs is non-trivial and is out of the scope of this paper, which is left as our future direction.

Table 7: Ablation study on the ViT backbone and pre-training datasets.

| Index | Backbone | Pre-training Datasets | Multi-scale Design | mAP on COCO ↑ | MPJPE on Human3.6M ↓ |
|-------|----------|----------------------|--------------------|---------------|----------------------|
| 1 | CPN | COCO | ✓ | 68.6 | 41.6 |
| 2 | HRNet-32 | COCO | ✓ | 74.4 | 41.4 |
| 3 | ViT | COCO | ✗ | 75.8 | 44.5 |
| 4 | ViT | COCO+AIC+MPII | ✗ | 77.1 | 41.9 |

## D   More Ablations on Backbones and Pre-training Datasets

We additionally conduct experiments on ViTPose [61], the recent state of the art on 2D HPE. The weights of ViTs [13] in ViTPose are initialized with MAE [19] pre-training, and then the model is further trained on 2D HPE datasets. The results are in Table 7, and observations are as follows.

**Backbone design outweighs the results on 2D HPE.** The first three rows in Table 7 demonstrate that better results on 2D HPE do not necessarily translate to better performance on 3D HPE: Although ViTPose performs best on COCO 2D HPE, it achieves worst results on Human3.6M. We attribute this result to the lack of multi-scale network design. ViTPose gains from powerful MAE pre-training with modern transformer architecture, while its network design is arguably simplified for 2D HPE. Specifically, ViTPose processes on tokenized image patches with transformers and finally increases the resolution of feature maps using 2D deconvolution layers. They use no multi-scale designs, *e.g.*, high-resolution feature branches or multi-scale feature fusion, as in CPN and HRNet. However, such techniques may help our approach learn more task-relevant information (*i.e.*, joint-context features) to localize joints in 3D.

**More pre-training datasets on 2D HPE improve the performance on the 3D Task.** A comparison between row 3 and row 4 in Table 7 shows that multi-dataset pre-training improves performance on both 2D HPE and 3D HPE. Plus, the gains on 3D HPE (5.8% error reduction) are even more significant than those on 2D HPE (1.3 points improvement on AP). We hypothesize that multi-dataset pre-training improves the generalization ability of learned backbone features. Therefore, the best performance on 2D HPE of ViTPose better transfers to the 3D task.

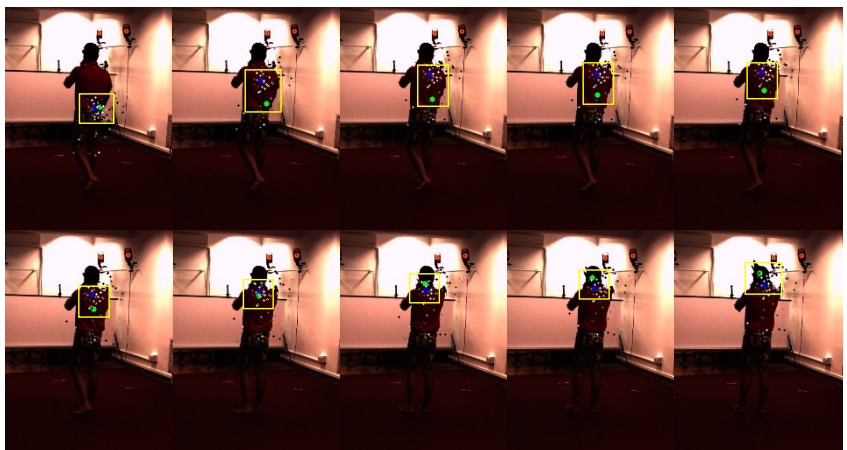

Figure 9: Visualization of consecutive frames on the Human3.6M test set where severe self-occlusion occurs. *Deformable Context Extraction* learns sampling points that attempt to discover ground truth joints given false 2D joint detection as reference.

## E   More Details and Visualization for Deformable Context Extraction

**More details.** The *Deformable Context Extraction* module extracts informative joint-centric context features from feature maps using deformable attention [75]. The sampling points of each attention head are initialized in different directions (w.r.t. the reference joint) to promote learning diverse

contextual clues from images. To prevent overly aggressive updates on sampling offsets (*e.g.*, they may go outside the image), the learning rate of linear layers that generate sampling offsets is set to 6.4e-4 (¹/₁₀ of that for other layers in the lifting model). We use 4 heads for deformable attention.

**Visualization of learned sampling points on consecutive video frames.** In Fig. 9, a subject in the Human3.6M test set raises his right arm where severe self-occlusion occurs, and the 2D pose detector fails to localize the right wrist (blue dots are detected results, and green dots are ground truth). We find that most sampling points (gray dots) are concentrated on the upper body of the subject. More interestingly, despite the unreliable 2D joint detection (reference points), some learned sampling points attempt to approach the ground truth. We indicate the sampling points that gain larger attention weights with higher brightness (we decrease the brightness of images for better visual effects). Note that we do *not* train sampling points using ground truth. This indicates that our adaptive context extraction strategy can learn informative contextual features based on the visual cues of images despite bad sampling references (*i.e.*, false joint detection), which helps reduce uncertainty brought by imperfect 2D pose detectors.

# F   More Visualization

We provide more qualitative results on Human3.6M (Fig. 10) and MPI-INF-3DHP (Fig. 11). Our single-frame method obtains reliable estimation in hard cases, *e.g.*, self-occlusion, and rare poses, compared to state-of-the-art multi-frame methods such as 351-frame MHFormer [29] and 81-frame P-STMO [50].

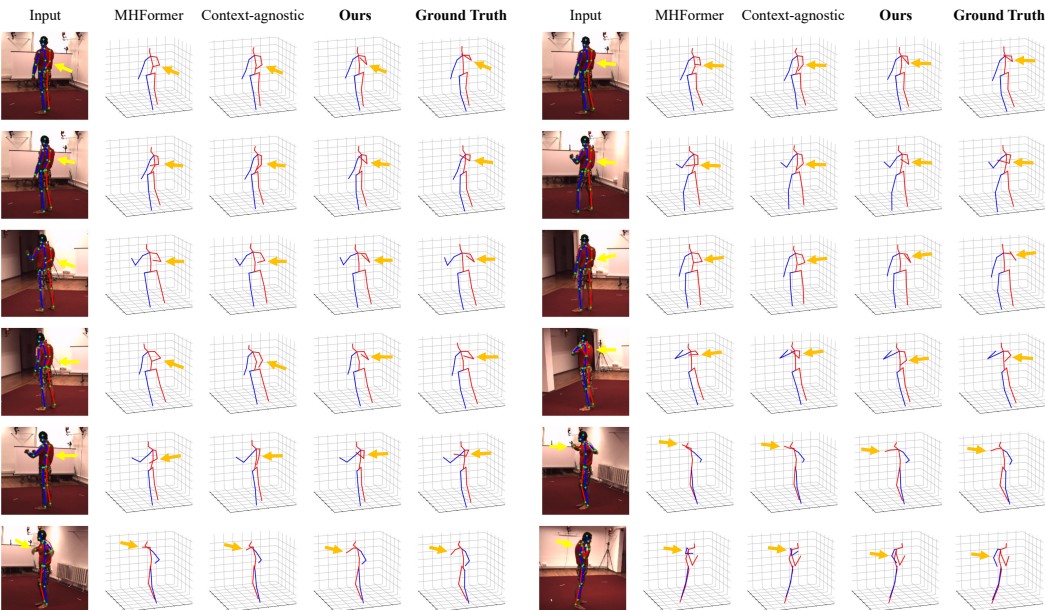

Figure 10: Qualitative comparison with MHFormer (351 frames) [29] and our context-agnostic counterpart (please refer to Sec. 4.3 for more details) on Human3.6M. Our method obtains reliable results despite severe self-occlusion, which may cause false 2D joint detection. Notable parts are indicated by arrows.

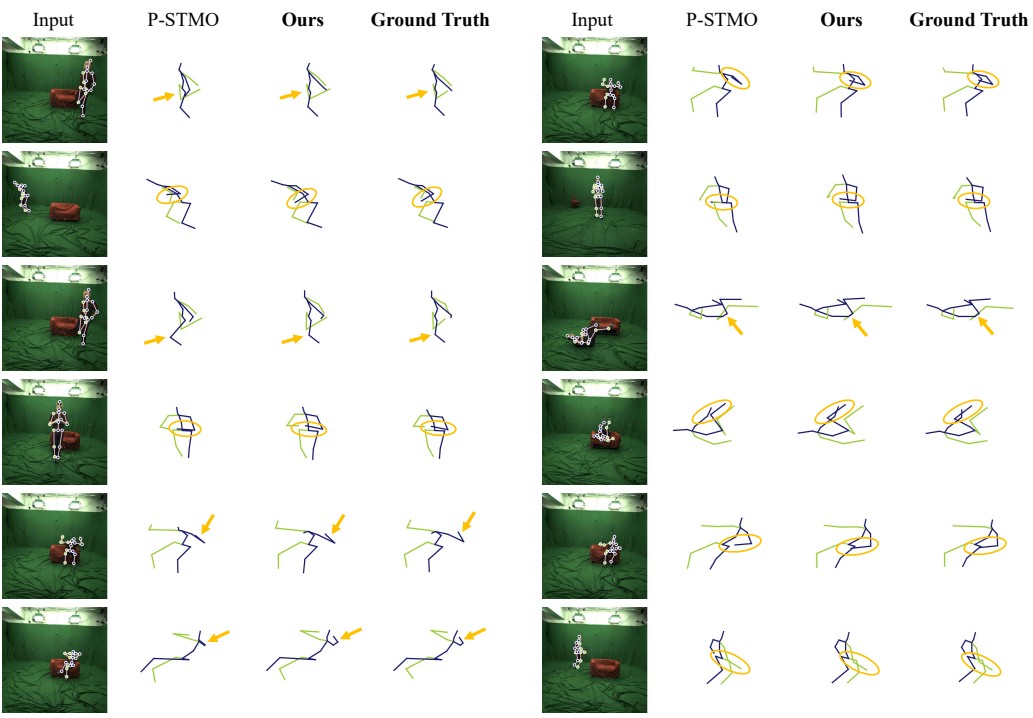

Figure 11: Qualitative comparison with P-STMO (81 frames) [50] on MPI-INF-3DHP. Our method infers correct results given rare poses (*e.g.*, the subject is lying on the ground and relaxing on the couch). Notable parts are indicated by arrows or circles.

