# OpenReview forum: "A Single 2D Pose with Context is Worth Hundreds for 3D Human Pose Estimation"
_NeurIPS.cc/2023/Conference — NeurIPS 2023 poster_

### Official Review · Reviewer_dcLQ · 2023-06-28

**Soundness:** 4 excellent
**Presentation:** 4 excellent
**Contribution:** 3 good
**Rating:** 7
**Confidence:** 5

**Summary:**

This paper proposes a context-aware method that combines the 2D image feature and the detected 2D key points for 3D human pose estimation. The joint-centric spatial context represented by the intermediate image feature is able to reduce the amguity in 3D lifting. Under this motivation, a novel framework that consists of a defrmable context extraction module, a pose-context feature fusion module and a spatial inter-joint modeling module is proposed. Experiment validates the effectiveness of the proposed modules in reducing calculation time and achieving better performance. On two standard benchmarks, the proposed method shows its effectiveness and stability without using any temporal information compared to other single-frame methods and multi-frame methods.

**Strengths:**

1. This paper is well presented and organized. The idea is clear and the method is somewhat novel.
2. New SOTA results are achieved with a significante improvement even compared to most multi-frame methods.
3. Experiments are well organized and the ablation study is insightful to support the claim of contributions.

**Weaknesses:**

1. Leveraging 2D intermediate feature is not a new idea. Some previous works also tried this route, such as [a][b][c]. Compared to other skeleton-based methods, either single-frame-based or multi-frame-based, utilizting 2D image feature will lead to more memory consumption. Such a kind of limitation needs to be considered.

[a] Yin, Binyi, et al. "Context-Aware Network for 3D Human Pose Estimation from Monocular RGB Image." 2019 International Joint Conference on Neural Networks (IJCNN). IEEE, 2019.

[b] Habibie, Ikhsanul, et al. "In the wild human pose estimation using explicit 2d features and intermediate 3d representations." Proceedings of the IEEE/CVF conference on computer vision and pattern recognition. 2019.

[c] Sun, Xiao, et al. "Integral human pose regression." Proceedings of the European conference on computer vision (ECCV). 2018.

**Questions:**

1. According to the study in Sec. 3 in the supplementary material, temporal information brings less improvement in the proposed method. Why the result when using 81 frames as PoseFormer is not compared? Could the authors give more explaination about this?

**Limitations:**

The authors have discussed their limitation of motion jitter without temporal information in the supplementary material. But the limitation of memory consumption should also be discussed as mentioned in the weakness.

---

> ### Author Rebuttal · Authors · 2023-08-07
>
> # Author Response to Reviewer dcLQ
> ## 1. Clarifications on Novelty
> We thank the reviewer for sharing related works, and we will discuss them in the `Related Work` section of the final version to make our work more complete. To address the reviewer's concern regarding novelty, we refer the reviewer to our `General Response`, where we comprehensively reaffirm the **novelty** and **contributions** of our approach. In addition, we would like to offer more clarifications on the difference between our method and the works [1,2,3] mentioned by the reviewer.
>
> While these works [1,2,3] also use image features, our work is *clearly different* for the reasons below:
>
> **1) Different Pipeline:** Our method follows the *two-stage* pipeline (2D pose detection then lifting, the dominant paradigm in recent years); Those works [1,2,3] estimate 3D human pose *end-to-end* from images *without explicitly introducing 2D human poses*. While using image features is **natural** even **stereotypical** for direct estimation (end-to-end) approaches, it is **non-trivial** to retrieve feature maps for lifting-based methods.
>
> **2) Different Feature Sources and Different Ways to Use Image Features:** In our work, we **reuse** feature maps from 2D pose detectors, which are **inherently** part of a lifting-based 3D HPE pipeline. Plus, we leverage feature maps **sparsely**, i.e., we extract and aggregate **point-wise feature vectors** based on the estimated 2D joints; Those works [1,2,3] introduce **independent** networks to extract image features and then use feature maps **as a whole** instead of in a sparse manner.
>
> **3) Different Research Focus:** We propose to leverage image features to remove the dependency of existing (two-stage) lifting-based methods on temporal information; Yin et al. [1] use image features to learn more accurate depth information; Habibie et al. [2] disentangle features for 2D pose from global image features to exploit large scale 2D pose datasets; Sun et al. [3] unify heatmap-based and regression-based methods, and learn 3D heatmaps from image features.
>
> [1] Context-Aware Network for 3D Human Pose Estimation from Monocular RGB Image. IJCNN 2019.
>
> [2] In the Wild Human Pose Estimation Using Explicit 2D Features and Intermediate 3D Representations. CVPR 2019.
>
> [3] Integral Human Pose Regression. ECCV 2018.
>
> ## 2. Memory Issue
>
> |      Method      |  Venue  | Frame Number | MPJPE$\downarrow$ (mm) | GPU Memory (MB) |
> |:----------------:|:-------:|:------------:|:----------------------:|:---------------:|
> |   MHFormer    | CVPR'22 |351|43.0|1651|
> |    MixSTE     | CVPR'22 |243|40.9|17305|
> | CA-PF-CPN (ours) | |1|41.6|721|
>
> We agree our method would consume more GPU memory than other lifting-based methods *when the frame number is kept the same*. However, we would like to mention that our approach does *not* cost more GPU memory compared to state-of-the-art multi-frame approaches that use heavy temporal modeling.
>
> We test the memory consumption of different methods during *stable training* using the same PyTorch code base on a single RTX 3090 GPU (24GB) with a batch size of 8. The results in the table above are from the command "torch.cuda.max_memory_allocated()" provided by PyTorch to return the peak allocated memory since the beginning of this program. The results show that our method costs less GPU memory than MHFormer and MixSTE by 2.3$\times$ and 24.0$\times$ respectively while achieving comparable performance.
>
> We thank the reviewer for mentioning this issue and will incorporate the results and analysis in our `Limitation` section.
>
> ## 3. Temporal Modeling
>
> |  Method | Frame Number | MPJPE$\downarrow$ | MPJVE$\downarrow$ |
> |:--:|:--:|:--:|:--:|
> | CA-PF-S |1|44.7|8.5|
> |    /    |3|44.2|4.8|
> |    /    |9|43.4|3.4|
> |    /    |27|40.2|2.1|
>
> **Note:** CA-PF-S is a *small* model variant. The last row (27 frames) is our new result.
>
> ### 3.1 Less Improvement?
> We additionally experiment on 27 frames, and the result is presented above (with previous results). We agree our temporal model gains less accuracy when using small frame numbers (e.g., 3 frames). However, more video frames bring significant improvements, e.g., our 27-frame model improves MPJPE and MPJVE by **7.8%** and  **38.2%** respectively over the 9-frame variant. We think the reason is that: Large human motions in long videos provide meaningful spatial varieties, and such temporal cues help reduce ambiguities in 2D-to-3D lifting. On the contrary, short video clips only include small motions while our joint-context features already encode such spatial changes (around joints). Therefore, improvements are minor when using small frame numbers.
>
> ### 3.2 Clarifications on Not Using 81 Frames
> In `Sec. 3 of the supplementary`, we extend our model to multi-frame settings **primarily aiming at solving the temporal consistency** issue (we discussed in `Supp. Sec. 2`), **not to pursue higher accuracy with more frames**. Important results from our experiments are listed below:
> 1. Even **short-term** temporal information helps reduce jitters well (only 3 video frames bring a 43.5% reduction in velocity error);
> 2. Our video-based model demonstrates better temporal consistency (and accuracy) than PoseFormer, e.g., our 9-frame model achieves less velocity error (MPJVE) than 9-frame PoseFormer (3.4 v.s. 4.8mm) and comparable results with 81-frame PoseFormer (3.4 v.s. 3.1mm);
> 3. Current experiments (up to 27 frames) demonstrate **consistent** improvements in both **accuracy** and **temporal smoothness**. We expect our model to further gain improvements using more input video frames.
>
> Given these results, the issue regarding temporal consistency should be well resolved by **short-term** temporal modeling, we think it **unnecessary** to further scale up the sequence length, which indeed **runs against our research motivation** (i.e., to reduce the reliance on temporal information).
>
> We are open to discussions if the reviewer has further concerns or questions!

---

> > ### Comment · Reviewer_dcLQ · 2023-08-18
> >
> > Thanks for the response. The memory issue I metioned should be compared to the single-frame-based methods if this method is presented as a single-frame pose estimation baseline. Although some points are overclaimed in the general response, I will keep my rating based on the strengths listed in my original review.

---

> > > ### Author Response · Authors · 2023-08-18
> > >
> > > We express our gratitude to the reviewer for the feedback! We will discuss the memory issue in our `Limitation` section following the suggestions from the reviewer. Additionally, we are committed to refining our paper to avoid potential overstatements.

---

### Official Review · Reviewer_jUmb · 2023-07-04

**Soundness:** 2 fair
**Presentation:** 3 good
**Contribution:** 2 fair
**Rating:** 4
**Confidence:** 5

**Summary:**

This paper proposes Context-Aware PoseFormer, the key idea is to extract multi-scale spatial features for 3D pose lifting from 2D pose.
The authors claim that the single-frame beats hundreds of frames and demonstrate that their single-frame methods achieve comparable or better results than multi-frame methods, such as poseformer.



**Strengths:**

The motivation of this article is clear, and the writing is simple and easy to understand.
The experiments in this paper verify the effect of the proposed method, verify that the spatial context information is very important, and the effect of the proposed context aware poseformer.
The ablation study in this article is detailed and verifies the effect of each module in the method.

**Weaknesses:**

I am skeptical of the claim in this paper that the single frame is better than the multiple frame method. This article only verifies that his single-frame method is better than some multi-frame methods such as poseformer, but there are 3 problems below：
1) The comparison of the methods in table 1 is unfair. For example, other methods use the 2D pose result of CPN as the output of the lifting network, while the method in this paper uses HRNet. It is well known that HRNet can perform better 2D detection than CPN.
2) In addition, the method in this paper also inputs more visual features from images, so this paper should also be compared with other methods that use visual features, such as IVT.
3) If the single-frame method in this paper works well under the same 2D network, it cannot be verified that it is definitely better than the multi-frame method. In the method of this article, is it possible to achieve better results by adding timing information? This is the consensus in the field, and this paper should also verify this.

In addition, the core contribution of the method in this paper is the multi-scale visual features of the image. The author uses a deformable method to extract multi-scale features to improve the performance of the network.
In the field of 3D posture, it is a consensus that extracting better spatial context features can improve performance.
In terms of context information extraction, this article has no additional contribution, only using other people's methods.
Therefore, in general, the method in this paper is a bit incremental in terms of innovation, and the contribution is not enough to be accepted by NeurIPS。

**Questions:**

see weakness

---

> ### Author Rebuttal · Authors · 2023-08-07
>
> # Author Response to Reviewer jUmb
> ## 1. Clarifications on Our Claim
> In this paper, we enable a single-frame approach to outperform multi-frame ones (e.g., that uses 351 frames) for the first time. We believe that strong experimental results (`Tab. 1` and `Tab. 2 in the main paper`) well verify the effectiveness of our method, and the reviewers (**THhA**,**ez57**,**4XvM**,**dcLQ**) also appreciate our results. We also include more recent papers for comparison. Please check the `Performance Comparison` section below!
>
> As new works are emerging in this field, we agree that we can *never* beat all multi-frame methods at any time, and therefore **we have *not* reached such a highly definitive conclusion**. However, we do **open up the *possibility* of significantly improving the performance of 3D HPE *in a highly disadvantaged position*** (i.e., with *no* access to temporal information), and the results we achieve are already encouraging and attractive (**4XvM**). We are open to suggestions from the reviewer to help us remove such confusion in the final version.
>
> ## 2. Performance Comparison
> ### 2.1 Fairness:
> We believe **comparing our CA-PF-CPN model variant with other state-of-the-art methods is fair**, as CPN is consistently used as the 2D pose detector (in `Tab. 1` we use one column to show the 2D pose detector used by different methods). In such a fair setting, our method (41.6mm, lower is better) outperforms a series of works, e.g., 81-frame PoseFormer [1] (44.3mm), 351-frame MHFormer [2] (43.0mm), and 243-frame P-STMO [3] (43.0mm), and achieves comparable results with 243-frame MixSTE [4] (40.9mm). We experiment with other backbones (e.g., HRNet) to primarily show the flexibility of our method to incorporate different 2D pose detectors. We will add more clarifications about the fairness of our experiments in `Sec. 4 of the main paper`.
> ### 2.2 Completeness:
> We had tried our best to gather both single-frame and multi-frame methods for comparison before CVPR 2023 papers were released (`Tab. 1 in the main paper`). Following the advice from review **ez57**, we compare with more recent SOTAs, including several CVPR 2023 and ICCV 2023 papers, and we will incorporate them in our final version. Please check our response to review **ez57**!
>
> [1] 3D Human Pose Estimation with Spatial and Temporal Transformers. ICCV 2021.
>
> [2] MHFormer: Multi-Hypothesis Transformer for 3D Human Pose Estimation. CVPR 2022.
>
> [3] P-STMO: Pre-Trained Spatial Temporal Many-to-One Model for 3D Human Pose Estimation. ECCV 2022.
>
> [4] MixSTE: Seq2seq Mixed Spatio-Temporal Encoder for 3D Human Pose Estimation in Video. CVPR 2022.
>
> ## 3. Comparison with IVT [5]
> We thank the reviewer for sharing such great work, and *we will include this paper in our `Related Work` section*. Our approach **principally differs** from IVT in the following aspects, although both methods use image features:
>
> **1) Motivation:** Our work aims at removing heavy temporal dependencies of existing lifting-based 3D HPE methods; IVT targets formulating video-based 3D HPE as an end-to-end learnable framework.
>
> **2) Task Pipeline:** We follow the dominant two-stage pipeline, which **explicitly uses 2D human poses as an intermediate representation**. We do **not** use any temporal information; IVT **directly** estimates 3D human poses from **videos** (2D human poses are **not** involved as an intermediate output, and **they use temporal information**).
>
> **3) Implementation Details:** We use input images of size 256 $\times$ 192, while IVT uses much larger images of size 512 $\times$ 512.
>
> Since IVT is **not open-sourced**, we are unable to conduct a direct comparison. However, more analysis and discussions regarding this work will be placed in the final version to make our work more complete!
>
> [5] IVT: An End-to-End Instance-guided Video Transformer for 3D Pose Estimation. ACM MM 2022.
>
> ## 4. Multi-frame Extension
>
> |  Method | Frame Number | MPJPE$\downarrow$ | MPJVE$\downarrow$ |
> |:-------:|:------------:|:-----------------:|:-----------------:|
> | CA-PF-S |       1      |        44.7       |        8.5        |
> |    /    |       3      |        44.2       |        4.8        |
> |    /    |       9      |        43.4       |        3.4        |
> |    /    |      27      |        40.2       |        2.1        |
>
> **Note:** CA-PF-S is a *smaller* model variant (than our full model) since temporal processing requires more computation and memory. The last row (27 frames) is our new result.
>
> This is an insightful question! We have already verified the flexibility of our method to extend to multi-frame settings in `Sec. 3 of the supplementary`! We move some results here with a new 27-frame result. The table above shows that our approach **gains consistently** in terms of **performance** (MPJPE, position error) and **temporal smoothness** (MPJVE, velocity error) using more video frames. Please check the supplementary for more details!
>
> ## 5. Concerns about Contribution
> We disagree that "the core contribution of our work method in this paper is the multi-scale visual features of the image." Leveraging visual representations from 2D pose detectors itself is a **non-trivial** and **novel idea** in the context of lifting-based 3D HPE, while our **novel framework** (**dcLQ**) to extract multi-scale joint-context features and fuse context features and joint embedding is **only one of our contributions**. Plus, we are the first to make a single-frame method outperform state-of-the-art multi-frame methods. We show our first attempt to tackle the heavy temporal reliance of existing lifting-based methods and establish strong single-frame baselines for future research.
>
> We hope the reviewer could kindly check our `General Response` to all reviewers, where we comprehensively elucidate the **novelty** and **contributions** of our method. We are open to discussions if the review has further questions or concerns.

---

> > ### Comment · Reviewer_jUmb · 2023-08-16
> >
> > Thanks for authors' feedback.
> >
> > First, the additional experiments solve the concerns on fair comparison and temporal information.
> > From the experiments, temporal information is also useful. Thus, I don't think the title of this paper is suitable, although their method outperforms other temporal methods.
> >
> > Second, the most important thing, as authors claimed that ''1. We Identify and Tackle A New Research Problem', ...'.
> > They claimed that they want to reduce the temporal information, but the additional experiments verified that temporal information could improve performance. Besides, if we do not use temporal information, the detected 3D poses are not smooth in temporal dimension, it's no practical in the application.
> > So I don't agree with the authors' claim about this.  I can agree that it's a good baseline compared with single-image methods. But I strongly disagree with the author's thesis on temporal motivation and contribution, which should be in single-frame approach track.
> >
> > Third, authors claimed that ''2. We Provide Fresh Insights regarding the Problem Cause ....''.
> > I also do not agree with this claim. Actually, there are many methods use visual features to improve the performance of 3D pose lifting.
> > It's overclaim.  I will provide some works later.
> > I am looking forward your response.

---

> > > ### Author Response · Authors · 2023-08-17
> > >
> > > ### Response to Reviewer jUmb [1/2]
> > >
> > > We thank the reviewer for the detailed feedback and are glad to address the concerns further!
> > >
> > > > First, the additional experiments solve the concerns on fair comparison and temporal information.
> > >
> > > We appreciate the reviewer for acknowledging that our performance comparison is fair and that our experiments on temporal modeling are solid.
> > >
> > > > From the experiments, temporal information is also useful. Thus, I don't think the title of this paper is suitable, although their method outperforms other temporal methods.
> > > >
> > > > Second, the most important thing, as authors claimed that ''1. We Identify and Tackle A New Research Problem', ...'. They claimed that they want to reduce the temporal information, but the additional experiments verified that temporal information could improve performance.
> > >
> > > We also appreciate the reviewer for acknowledging the superiority of our single-frame method over other multi-frame methods.
> > >
> > > We wish to emphasize that the assertion **"temporal information still helps improve performance"** does *not* contradict the fact that **"our approach achieves strong performance without using temporal modeling."** It is important to clarify that we have *never* negated the significance of temporal information. In fact, the enhancements derived from temporal information underscore the **versatility** and **scalability** of our method.
> > >
> > > Our title endeavors to emphasize a new **alternative** to the conventional approach of incorporating an increased number of video frames, consequently enhancing the accuracy of 3D human pose estimation. While we acknowledge that our title might sound overly assertive, we are receptive to refining it. What is the reviewer's perspective on the title **"Single 2D Pose with Context is Worth Hundreds of Frames for 3D Human Pose Estimation"**? We welcome the reviewer's insights and recommendations.
> > >
> > > > Besides, if we do not use temporal information, the detected 3D poses are not smooth in temporal dimension, it's no practical in the application. So I don't agree with the authors' claim about this.
> > >
> > > We agree that temporal smoothness is a limitation of our method. **It's important to note that temporal smoothness is a general challenge faced by all single-frame methods in contrast to multi-frame methods, as non-temporal approaches inherently struggle to ensure such smoothness.** Thus, it's reasonable to assert that our single-frame method should not be critiqued on this particular dimension.
> > >
> > > Nevertheless, it's worth highlighting that in `Figure 5, Section 4.4`, our approach demonstrates improved temporal smoothness when compared against the context-agnostic baseline, despite not having direct access to temporal information.
> > >
> > > In  `Tab. 1 of the supplementary`, we also provide a solution to solve such limitation by extending our method to model **short-term** temporal dependencies. In terms of the MPJVE metric (where lower values indicate better smoothness), our 9-frame temporal model achieves comparable results with 81-frame PoseFormer (3.4 v.s. 3.1mm); our 27-frame model even significantly outperforms 81-frame PoseFormer (2.1 v.s. 3.1mm). These experimental findings underscore that **our approach reduces the necessity for extensive long-term temporal modeling to achieve satisfactory temporal smoothness, distinguishing it from conventional multi-frame methods**.
> > >
> > > Consequently, the introduction of short-term temporal extensions to our model (e.g., utilizing 9 video frames) holds real-world applicability in mitigating smoothness concerns.

---

> > > > ### Author Response · Authors · 2023-08-17
> > > >
> > > > ### Response to Reviewer jUmb [2/2]
> > > >
> > > > > I can agree that it's a good baseline compared with single-image methods. But I strongly disagree with the author's thesis on temporal motivation and contribution, which should be in single-frame approach track.
> > > >
> > > > We further thank the reviewer for appreciating that our work establishes a good single-frame baseline.
> > > >
> > > > It seems there could be a contradiction within the reviewer's feedback. Reviewer **jUmb** initially praised our work for its transparent motivation and accessible writing ("The motivation of this article is clear, and the writing is simple and easy to understand"). Similarly, reviewer **ez57** also offered positive feedback on our motivation ("The motivation and observation are clear and meaningful"). However, the subsequent comments by reviewer **jUmb** appear to raise doubts about our motivation.
> > > >
> > > > Our work is motivated by several challenges posed by the adoption of extensive long-term temporal modeling, a default setting in 3D HPE: high computation, performance saturation, and the non-causal problem. This work presents an alternative approach to long-term temporal modeling while showing encouraging performance. Specifically, our CA-PF-CPN model surpasses the prior SOTA single-frame method [1] by an impressive **13.2%**, and it achieves comparable, if not superior, results in comparison to leading multi-frame methods that leverage up to 351 video frames. Consequently, **we believe our contribution would be significantly underrated if our work is simply discussed in the context of non-temporal methods**.
> > > >
> > > > [1] Learning skeletal graph neural networks for hard 3d pose estimation. ICCV 2021.
> > > >
> > > > > Third, authors claimed that ''2. We Provide Fresh Insights regarding the Problem Cause ....''. I also do not agree with this claim. Actually, there are many methods use visual features to improve the performance of 3D pose lifting. It's overclaim. I will provide some works later. I am looking forward your response.
> > > >
> > > > Based on our observation: the majority of current SOTA lifting-based methods discard image features to accommodate extremely long input 2D pose sequences (e.g., 243 frames for MixSTE [2] and P-STMO [3], 351 frames for MHFormer [4]). In contrast, we present an alternative approach that harnesses discarded features to enhance performance.
> > > >
> > > > [2] Mixste: Seq2seq mixed spatio-temporal encoder for 3d human pose estimation in video. CVPR 2022.
> > > >
> > > > [3] P-stmo: Pre-trained spatial temporal many-to-one model for 3d human pose estimation. ECCV 2022.
> > > >
> > > > [4] Mhformer: Multi-hypothesis transformer for 3d human pose estimation. CVPR 2022.
> > > >
> > > > While some methods also incorporate image features [5,6,7], our work stands out due to the following distinctions:
> > > >
> > > > **1) Different Motivation:** We leverage image features to remove the dependency of existing (two-stage) lifting-based methods on temporal information; None of these studies [5,6,7] have employed image features for this specific purpose.
> > > >
> > > > **2) Different Pipeline:** Our method follows the 2D-to-3D lifting pipeline (generating 2D pose by the 2D pose detector first); Importantly, the 2D pose detector is frozen to ensure consistent 2D pose outputs. **We do not finetune the 2D pose detector during training!** We **reuse** the discarded feature maps, which are **inherently** part of a lifting-based 3D HPE pipeline.
> > > >
> > > > In contrast, existing methods [5,6,7] introduce **extra** **learnable** image backbones and **no longer explicitly use 2D poses**. Consequently, their usage of image features becomes part of an end-to-end process (estimating the 3D pose directly from images), deviating from the strict 2D-to-3D lifting pipeline since the 2D pose is no longer present.
> > > >
> > > > **3) Different Ways to Use Image Features:** In our work, we reuse feature maps from 2D pose detectors, **employing a sparse strategy**, i.e., we extract and aggregate point-wise feature vectors based on the estimated 2D joints (to make the features extracted more informative). These works [5,6,7] use image feature maps **as a whole** instead of in a sparse manner. This distinction makes our method lightweight yet capable of achieving superior performance compared to those methods.
> > > >
> > > > [5] Context-Aware Network for 3D Human Pose Estimation from Monocular RGB Image. IJCNN 2019.
> > > >
> > > > [6] In the Wild Human Pose Estimation Using Explicit 2D Features and Intermediate 3D Representations. CVPR 2019.
> > > >
> > > > [7] Integral Human Pose Regression. ECCV 2018.
> > > >
> > > > We are glad and open to discussions if the reviewer could share more such works with us!

---

### Official Review · Reviewer_4XvM · 2023-07-05

**Soundness:** 3 good
**Presentation:** 3 good
**Contribution:** 3 good
**Rating:** 5
**Confidence:** 4

**Summary:**

The paper targets the challenging 3D pose estimation problem based on a new context-aware lifting algorithm. The proposed approach is simple to implement and reproduce. Attractive experimental results have been reported on the challenging benchmarks. Also, the detailed ablations well validate the design of the proposed algorithm.

**Strengths:**

1. The idea of lifting 2d image features for 3D human pose estimation is interesing.
2. The proposed algorithm report state-of-art performance on the 3d pose estimation benchmark.
3. It also provides sufficient ablation studies to validate the algorithm design of the proposed approach.

**Weaknesses:**

1. In stage 2, it involves a Deformable context extraction module to extract the context features (F1, F2, F3, P). How is the feature difference against H1-H3? Also, is it possible to add H1-H3 with positional embeding to the pose-context feature fusion module?

2. It would be reasonable that 2D pose information can be helpful for the 3d pose estimation. But lifting from 2d to 3d is not-trivial. How could the proposed approach guarantee the consistency with the 3d location of the pose?

3. If there are more temporal frames available, is it possible to further boost the performance of the proposed approach?

**Questions:**

Please address the questions in the weakness section.

**Limitations:**

The paper does not discuss the potential limitations of the proposed algorithm.  I would suggest the authors to have a discussion of the limitations (like the cases with high self-occlusion which temporal information may be helpful).

---

> ### Author Rebuttal · Authors · 2023-08-07
>
> # Author Response to Reviewer 4XvM
> ## 1. Details about Deformable Context Extraction
> $H_1$-$H_3$ refer to the *raw feature maps* produced by 2D pose detectors, while $F_1$-$F_3$ are *extracted feature vectors* named "Context Features" from the corresponding feature maps using *Deformable Context Extraction* (DCE). Specifically, we initialize $F_1$-$F_3$ as the feature vectors directly sampled at the detected 2D joints from $H_1$-$H_3$. As the detected joints inevitably introduce noise, e.g., due to occlusions, we use deformable attention [1] to produce a small set of sampling points around each detected joint and fuse their features. We provide a visualization of sampled points in `Fig. 4 of the main paper`. Such sampling-and-feature-mixture process happens for N1 (the layer number of DCE) times, and the final output of DCE that represents the context features for each joint from each feature map is denoted by $F_1$-$F_3$. $P$ is the linear embedding of 2D joint coordinates. Please check `Fig. 3` and `Sec. 3 (L173-200) in the main paper` and `Sec. 5 in the supplementary` for more details and visualization!
>
> [1] Deformable DETR: Deformable Transformers for End-to-End Object Detection. ICLR 2021.
>
> ## 2. Add H1-H3 to the Pose-Context Feature Fusion module
> *Pose-Context Feature Fusion* primarily aims at fusing image cues (joint-context features, $F_1$-$F_3$) and position cues (joint-coordinate embedding, $P$) for each joint using transformers. Adding $H_1$-$H_3$ to *Pose-Context Feature Fusion* is possible and may provide more improvements as they intuitively encode rich context features. However, some concerns should be addressed:
> 1. Processing feature maps ($H_1$-$H_3$) generally costs more GPU memory and computation compared to vectorized features ($F_1$-$F_3$) since $H_1$-$H_3$ also contain features for uninformative background;
> 2. While it is convenient for transformers to process feature vectors ($P$, $F_1$-$F_3$) with the same dimension, it is not straightforward to process feature vectors ($P$) and feature maps ($H_1$-$H_3$) simultaneously as they have different shapes. We may need extra model design to make such a process possible.
>
> Given the concerns above, we prefer to use vectorized features ($F_1$-$F_3$) rather than feature maps ($H_1$-$H_3$). Since we extract $F_1$-$F_3$ from $H_1$-$H_3$ based on detected 2D joints, they are reasonably informative to provide desirable joint-context features. We thank the reviewer for such an enlightening question, and we will incorporate the analysis and explanations about module design in our `Method` section in the final version.
>
> ## 3. How to ensure 2D-to-3D consistency?
> This is **not** a weakness of our method! We follow the *common practice* in the domain: the format (e.g., joint number and type) of 2D joints and 3D joints (input-output pairs) are pre-defined and aligned before training. Then our network learns to lift 2D joints to 3D via supervised learning using paired 2D and 3D data.
>
> ## 4. Multi-frame Extension
>
> |  Method | Frame Number | MPJPE$\downarrow$ | MPJVE$\downarrow$ |
> |:-------:|:------------:|:-----------------:|:-----------------:|
> | CA-PF-S |       1      |        44.7       |        8.5        |
> |    /    |       3      |        44.2       |        4.8        |
> |    /    |       9      |        43.4       |        3.4        |
> |    /    |      27      |        40.2       |        2.1        |
>
> **Note:** CA-PF-S is a small model variant since temporal processing requires more computation and memory. The last row (27 frames) is our new result.
>
> In `Sec.3 of the supplementary`, we have described a simple way to extend our single-frame approach to model temporal dependencies. Here we move some results from `Supp. Tab. 1` with a **new** result using 27 frames, showing that by using more video frames, our method *gains consistently* in terms of *performance* (position error, MPJPE) and *temporal smoothness* (velocity error, MPJVE). Given the results above, we expect our model to improve further if we increase the frame number.
>
> ## 5. Discussion on Limitation
> Indeed, we **have included** the discussion on the limitation of our method in `Supp. Sec. 2`. Please check! Besides, we agree that temporal information helps to resolve self-occlusion by providing features from unoccluded video frames. Interestingly, we find that *contextual features may also improve the robustness of the model to self-occlusion*. In `Fig. 5 of the supplementary`, we provide two video clips where strong self-occlusion (due to pose or clothing) makes the 2D human pose unreliable. Despite inaccurate 2D input, our method gives robust results since we also leverage spatial contextual clues from images to localize joints in 3D in addition to noisy 2D joints.
>
> We think "how our model deals with self-occlusion and how temporal information helps model" would provide exciting insights, and we will include more analysis in the final version. Plus, we will move the `Limitation` section to the main paper. We thank the reviewer for such an insightful suggestion.

---

> > ### Comment · Reviewer_4XvM · 2023-08-14
> >
> > The rebuttal well addressed most of my concerns.

---

> > > ### Author Response · Authors · 2023-08-14
> > >
> > > We appreciate your feedback! If you have any more concerns, we're more than happy to assist in addressing them.

---

### Official Review · Reviewer_ez57 · 2023-07-05

**Soundness:** 4 excellent
**Presentation:** 3 good
**Contribution:** 3 good
**Rating:** 7
**Confidence:** 5

**Summary:**

This paper leverages the readily available intermediate visual representations for 3d human pose estimation. The method discards temporal information to solve the time-intensive issue of existing lifting-based methods. The authors design a simple pipline, named Context-Aware PoseFormer, to extract informative context features and fuse these features to learn more positional clues.

After reading authors' rebuttal, my previous concerns were all addressed.

**Strengths:**

1.The paper is generally well written. The problem to address and the shortcomings of the existing approaches are discussed well.

2. The motivation and observation are clear and meaningful, and the network is well designed to solve these problems.

3.The experiments demonstrate well the superiority of the proposed method over the prior art. The authors also provide adequate ablation studies.

**Weaknesses:**

More STOA methods in 2023 should be chosen for comparison.

**Questions:**

1. Does the resolution of input images influence the performance?

2. It would be interesting to apply the proposed method to other related tasks like hand pose estimation to verify its generalization capability.

**Limitations:**

The authors should describe the limitation of the proposed method.

---

> ### Author Rebuttal · Authors · 2023-08-07
>
> # Author Response to Reviewer ez57
> ## 1. More Comparisons with SOTA in 2023
> We thank the reviewer for the advice on performance comparison! We agree that comparing with more papers would make our work more complete. We include recently released CVPR'23 and ICCV'23 papers in the table below and will incorporate the results in our final version.
>
> |         Method        |   Venue  | 2D Pose Detector | Frame Number | MPJPE on Human3.6M$\downarrow$ (mm) |
> |:---------------------:|:--------:|:----------------:|:------------:|:-----------------------------------:|
> | MPM [1]               | arXiv'23 |        CPN       |      243     |                 42.3                |
> | STCFormer [2]         |  CVPR'23 |        CPN       |      243     |                 41.0                |
> | GLA-GCN [3]           |  ICCV'23 |        CPN       |      243     |                 44.4                |
> | CA-PF-CPN (ours)      |          |        CPN       |       1      |                 41.6                |
>
> [1] MPM: A Unified 2D-3D Human Pose Representation via Masked Pose Modeling. arXiv 2023.
>
> [2] 3D Human Pose Estimation with Spatio-Temporal Criss-cross Attention. CVPR 2023.
>
> [3] GLA-GCN: Global-local Adaptive Graph Convolutional Network for 3D Human Pose Estimation from Monocular Video. ICCV 2023.
>
> All approaches listed above (except ours) use **243** video frames, while we only use a **single** frame. Compared to the latest SOTAs in 2023, our approach is still highly competitive: Our CA-PF-CPN outperforms MPM [1] (arXiv'23) and GLA-GCN [3] (ICCV'23) by 1.7% and 6.8% respectively and achieves comparable performance with STCFormer [2] (CVPR'23). Our performance can be further improved using larger backbones (`Tab. 1, main paper`) or temporal information (`Sec. 3, supplementary`).
>
> ## 2. The Impact of Image Resolution
> Thanks for such an insightful question! We think it is exciting and meaningful to explore the impact of image resolution on our method. Our paper uses images of size 256 $\times$ 192, a standard setting in COCO 2D HPE. As 384 $\times$ 288 is also widely used for input image size in 2D HPE, we experiment on our CA-PF-HRNet-32 with such resolution, and the results are shown below.
>
> |         Method        |    Image Size    | GFLOPs of Backbone | MPJPE$\downarrow$ (mm) |
> |:---------------------:|:----------------:|:------------------:|:----------------------:|
> | CA-PF-HRNet-32 (ours) | 256 $\times$ 192 |         7.1        |          41.4          |
> |           /           | 384 $\times$ 288 |        16.0        |          39.3          |
>
> Increasing the image size from 256 $\times$ 192 to 384 $\times$ 288 brings a 5.1% error reduction. While the model architecture and hidden dimensions are unchanged for the backbone network, it produces larger feature maps. Larger feature maps encode more fine-grained joint-context features and therefore improving performance. This result is consistent with our finding in `Tab. 4 of the main paper`, showing that high-resolution features contribute more than low-resolution ones. We will incorporate the result and analysis in our `Abation Study (Sec. 4.3, main paper)`.
>
> ## 3. Generalization Ability to Related Fields
> We apologize for being unable to show the results at present, as our computational resources are limited. Exploring the generalization ability of our approach to other domains will definitely be our future direction! For example, we expect our method to work on human mesh reconstruction, hand pose estimation, and, more generally, other 3D tasks where the 2D skeleton representation of the target 3D object is readily available. We believe leveraging the well-learned 2D features that produce such skeletons will reduce ambiguities in 2D-to-3D lifting and significantly improve the performance of the 3D task. We will open a `Future Work` section in the final version to discuss potential directions.
>
> ## 4. Discussion on Limitation
> In the `supplementary`, we have included the discussion on the limitation of our method in `Sec. 2` and our corresponding solution in `Sec. 3`. Please check!

---

> > ### Comment · Reviewer_ez57 · 2023-08-18
> >
> > I have no further questions. I recommend to accept this paper.

---

> > > ### Author Response · Authors · 2023-08-18
> > >
> > > We thank the reviewer again for the considerate feedback and valuable insights! We will integrate the experimental results and analysis to enhance the comprehensiveness of our paper.

---

### Official Review · Reviewer_THhA · 2023-07-06

**Soundness:** 3 good
**Presentation:** 3 good
**Contribution:** 3 good
**Rating:** 5
**Confidence:** 4

**Summary:**

This paper introduces Context-Aware PoseFormer, a new approach that leverages intermediate visual representations from pre-trained 2D pose detectors to implicitly encode spatial context for 3D human pose estimation. Despite the simple network structure, the proposed method outperforms existing state-of-the-art single- and multi-frame methods by a large margin without relying on temporal information.

**Strengths:**

-This paper provides an interesting insight that leverages the spatial contextual information of pre-trained 2D pose detectors, referred to as "context-aware" information, to improve the accuracy of 3D human pose estimation.

- The proposed approach provides considerable benefits by elevating the accuracy of single-frame methods to a level on par with state-of-the-art multi-frame methods

**Weaknesses:**

-Incremental novelty in the idea of context aware features and lack of discussion on the existing context aware features.
   - Modeling context-aware operations or networks has been studied in many other computer vision topics. For example, pixel-aligned features [1, 2] have been extensively used in 3D human/hand reconstruction from single/multi images. This pixel-aligned feature is demonstrated to be the most useful module in Table 3 while other modules only improve the performance marginally.

- Incomplete evaluation regarding the pretrained context-aware feature network.
    - For the pretraining, what's the impact of different pre-training datasets (e.g. number of training data) on the estimation?
    -  For the pretraining backbone, what's the impact of latest backbone on the final results?  e.g. ViT-based [3] pre-trained backbones MAE [4].

Reference
- [1] Pymaf: 3d human pose and shape regression with pyramidal mesh alignment feedback loop. ICCV 2021
- [2] Pifu: Pixel-aligned implicit function for high-resolution clothed human digitization. ICCV2019
- [3] An Image is Worth 16x16 Words: Transformers for Image Recognition at Scale ICLR
- [4] Masked autoencoders are scalable vision learners. CVPR 2022





**Questions:**

Please see the weakness

**Limitations:**

Please see the weakness

---

> ### Author Rebuttal · Authors · 2023-08-07
>
> # Author Response to Reviewer THhA
> ## 1. Clarifications on Novelty
> We disagree that the novelty of our idea is incremental. We hope the reviewer could check our `General Response` for more comprehensive clarifications on the novelty of our work.
>
> We thank the reviewer for sharing related works that leverage pixel-aligned features [1,2], and we will discuss them in our `Related Work (Sec. 2)` in the final version! However, our method can be distinctively differentiated from them in the following aspects:
>
> **1) Motivation:** Pixel-aligned features [1,2] were primarily proposed to **reduce the misalignment** between estimated 3D representations and input 2D images. However, we propose to leverage joint-context features to **reduce ambiguities** in lifting-based 3D HPE, e.g., depth ambiguity, thus removing the need for long-term temporal processing.
>
> **2) Source of Feature Maps and the Way to Extract Features of Interest:** Even though our works use feature maps in a point-wise manner similar to pixel-aligned features [1,2], technical details differ significantly. First, while PyMAF [1] and PIFu [2] introduce an **independent** image encoder to produce image features, we **reuse** the features maps from 2D pose detectors, which are *inherently* part of the two-stage 3D HPE pipeline. Plus, we extract joint-context features based only on **estimated 2D joints**. On the contrary, pixel-aligned features [1,2] are extracted by **projecting estimated 3D** meshes [1] or query 3D points [2] on the image plane with camera parameters to enforce the consistency between 2D input and 3D estimation. We do not use such 3D-to-2D projection.
>
> **3) Different Research Task and Pipeline:** Given input images, PyMAF [1] reconstructs human *meshes*, and PIFu [2] learns an implicit function for human *surfaces* and *textures*. In contrast, our work focuses on lifting-based 3D HPE, namely inferring 3D *joint coordinates* from images where 2D human joints are intermediate representations. PyMAF [1] and PIFu [2] do *not* use 2D human joints.
>
> [1] Pymaf: 3d human pose and shape regression with pyramidal mesh alignment feedback loop. ICCV 2021.
>
> [2] Pifu: Pixel-aligned implicit function for high-resolution clothed human digitization. ICCV 2019.
>
> ## 2. Importance of Different Modules
> We agree in Tab.3 joint-context features bring the most significant improvements over other modules. However, *Pose-Context Feature Fusion* and *Deformable Context Extraction* also provide *non-trivial* gains (**1.6%** and **3.3%** respectively). We want to highlight the **difficulty** of such gains from the two modules:
> 1. The two modules build on an **already-competitive** baseline (43.5mm on Human3.6M, lower is better), which is comparable to many state-of-the-art methods, e.g., 351-frame MHFormer (43.0mm, CVPR'22) and 243-frame P-STMO (43.0mm, ECCV'22);
> 2. They bring an error reduction of  2.1mm (from 43.5 to 41.4mm) together, while MHFormer (CVPR'22) and P-STMO (ECCV'22) only outperform the prior art PoseFormer (44.3mm, ICCV'21) by 1.3mm.
>
> ## 3. More Evaluation
> We also thank the reviewer for the advice on more evaluation regarding the pre-trained backbone networks. **We have already conducted experiments with commonly used 2D pose detectors**, e.g., SimpleBaseline (ResNet-50), CPN, and HRNet, on two pre-training tasks, i.e., (the most popular) COCO 2D HPE and ImageNet classification. Such comparisons can be found in the `main paper (Sec. 4.1)` and `supplementary material (Sec. 6)`.
>
> We additionally conduct experiments on ViTPose [3], the recent SOTA on 2D HPE. The weights of ViTs in ViTPose were initialized with MAE [4] pre-training, then the model was further trained on 2D HPE datasets. The results and observations are summarized below.
>
> [3] ViTPose: Simple Vision Transformer Baselines for Human Pose Estimation. NeurIPS 2022.
>
> [4] Masked autoencoders are scalable vision learners. CVPR 2022.
>
> | Index | Backbone | Pre-training Datasets | Multi-scale Design | mAP on COCO$\uparrow$ | MPJPE on Human3.6M$\downarrow$ (mm) |
> |:--:|:--:|:--:|:--:|:--:|:--:|
> |1|CPN|COCO|Y|68.6|41.6|
> |2| HRNet-32 |COCO|Y|74.4|41.4|
> |3|ViT [3]|COCO|N|75.8|44.5|
> |4|ViT [3]| COCO+AIC+MPII |N|77.1|41.9|
>
> **Note:** CPN and HRNet do *not* provide official multi-dataset pre-trained weights, so we apply this setting to ViTPose.
>
> ### 3.1 The Impact of Backbone Design
> **Backbone Design Outweighs the Results on 2D HPE.** The first three rows in the table above demonstrate that better results on 2D HPE do *not* necessarily translate to better performance on 3D HPE: Although ViTPose performs best on COCO 2D HPE, it achieves worst results on Human3.6M. We attribute this result to the lack of multi-scale network design. ViTPose gains from powerful MAE pre-training with modern transformer architecture, while its network design is arguably simplified for 2D HPE. Specifically, ViTPose processes on tokenized image patches with transformers and finally increases the resolution of feature maps using 2D deconvolution layers. They use *no* multi-scale designs, e.g., high-resolution feature branches or multi-scale feature fusion, as in CPN and HRNet. However, such techniques may help our approach learn more task-relevant information (i.e., joint-context features) to localize joints in 3D.
>
> ### 3.2 The Impact of Pre-training Datasets
> **More Pre-training Datasets on 2D HPE Improve the Performance on the 3D Task.** A comparison between row 3 and row 4 in the table above shows that multi-dataset pre-training improves performance on both 2D HPE and 3D HPE. Plus, the gains on 3D HPE (5.8% error reduction) are even more significant than those on 2D HPE (1.3 points improvement on AP). We hypothesize that multi-dataset pre-training improves the generalization ability of learned backbone features. Therefore, the best performance on 2D HPE of ViTPose better transfers to the 3D task.
>
> We find these results inspiring and will incorporate them in `Sec. 4` of the final version.

---

> ### Author Response · Authors · 2023-08-20
>
> We kindly wish to remind the reviewer to consider our response and additional experiments. We are more than willing to provide further clarifications if there are any lingering questions or concerns.

---

### Author Rebuttal · Authors · 2023-08-07

# General Response
We thank the reviewers for their careful reading and considerate feedback, and we are thrilled to receive the rating of 4 5 5 7 7!

We are glad that reviewers unanimously agree that *Context-Aware PoseFormer* is a **simple but effective approach** demonstrated by **strong experimental results** ("considerable benefits" (THhA), "superiority over the prior art" (ez57), "attractive experimental results" (4XvM), "experiments verify the effect of the proposed method" (jUmb), "new SOTA results" (dcLQ)). We’re further glad that reviewers agree that our **idea is interesting** ("interesting insight" (THhA), "idea is interesting" (4XvM)) and that our **motivation is well illustrated** with **clear writing** ("motivation and observation are clear and meaningful" (ez57), "motivation is clear" and "easy to understand" (jUmb), "paper is well presented and organized," and "idea is clear" (dcLQ)). The reviewers also agree that our ablations are "adequate" (ez57), "detailed" (jUmb), "insightful" (dcLQ), and "well validate the design of the proposed algorithm" (4XvM).

However, some reviewers raised their concerns regarding the novelty of our approach. To address those concerns, we first reaffirm the **novelty** and **contribution** of the proposed method, then answer specific questions for each reviewer in the corresponding rebuttal space, and we will incorporate all feedback in the final version.

## 1. We Identify and Tackle A **New Research Problem**
The dominant paradigm in 3D human pose estimation (HPE) literature is lifting a 2D joint sequence to 3D (dubbed as *lifting-based* methods, in contrast to direct estimation from images). Heavily using temporal information (with up to 351 video frames) has been a *default* setting in the field and has proved to boost performance. However, we point out in this paper that such reliance on temporal information brings several problems: high computation, performance saturation, and the non-causal issue. Despite the problems, **reducing the heavy reliance on temporal modeling in lifting-based 3D HPE has not yet been visited. We for the first time enable a single-frame method to outperform state-of-the-art multi-frame methods that even use hundreds of video frames.**

### Broader Impacts on the Community
Learning more powerful spatial-temporal representations with ever-increasing video frames (up to 351) has long been the research focus in lifting-based 3D HPE. However, **our approach breaks through such common practice (i.e., using no temporal information) while achieving encouragingly strong performance**. We establish solid baselines for the community and hope our work inspires more research to **think out of the box**: large-scale spatial-temporal modeling is not the only way to improve 3D HPE. We also believe that a research community should embrace different approaches, and our method can be such a starting point.

## 2. We Provide **Fresh Insights** regarding the Problem Cause
We discover the fundamental cause of existing methods' heavy temporal reliance by revisiting the dominant two-stage pipeline in the literature (L42-56). We point out that the 2D skeleton alone is insufficient to deal with ambiguities in 2D-to-3D lifting, making previous works resort to long-term temporal clues to mitigate ambiguities. To address this problem, we propose to retrieve the *discarded* visual representations (intermediate feature maps) learned in the 2D pose detection stage, as such representations encode visual cues from images that potentially help to reduce ambiguities. These insights are **fresh** in the domain and appreciated by reviewer **THhA**.

## 3. We Design a **Novel Framework** to Solve the Problem
"How to leverage the visual representations learned by 2D pose detectors" is a **non-trivial** question. Naively incorporating the global image features into the lifting process may introduce unnecessary memory and computational costs on the background. We comprehensively consider the task pipeline and design a novel approach to extract most task-relevant information, named *joint-context features* from the feature maps with detected 2D joints as a reference. This approach enables us to attend to the most informative regions, i.e., the joints of our interest. Then we fuse the extracted context features with joint coordinate embedding using transformers. **The collaboration of two stages (2D HPE and 2D-to-3D lifting) and the approach to extract joint-context features from images are novel in lifting-based 3D HPE** ("novel framework" (**dcLQ**)).

### While Using Image Features is Not New, What Makes Our Approach Different?
Although image features may have been explored in related fields, it does not indicate a lack of novelty in our work. Leveraging feature maps from 2D pose detectors is **non-trivial** in the context of lifting-based 3D HPE, as lifting only 2D joints to 3D is the *default* pipeline. **Our idea's uniqueness shines through its thorough understanding of the task setting.** We derive solutions directly from the task pipeline (i.e., leveraging readily available feature maps based on detected 2D joints, all elements are *off-the-shelf* in the pipeline), eliminating the need for external interventions (e.g., without introducing an extra network to extract image features). We believe it is important to evaluate the novelty of a research idea based on the domain challenge, as different domains may face different challenges.

## 4. Other Merits: Our Method is **Easy** and **Simple**

1) We *reuse* 2D pose detector feature maps instead of introducing extra networks, avoiding potentially high computational overheads;

2) 2D pose detectors are only pre-trained with 2D HPE (i.e., such backbones are largely available), requiring no finetuning on the 3D task or multi-stage training, which eases the training pipeline;

3) Our method is compatible with different 2D pose detectors;

4) Our framework is simple (**THhA**,**ez57**,**4XvM**).

---

### Decision · Program_Chairs · 2023-09-21

**Decision:**

Accept (poster)

**Comment:**

This paper received four positive ratings and one negative rating. The main concerns raised by the negative reviewer is that the paper mis-claimed the advantages over video-based methods. I agree with the reviewer that this should be revised in the camera ready. However, this does not seem a serious issue and as promised by the authors that they will revise the claims. Other than that, the paper presents an interesting approach which combines image features with 2D joints for 3D pose estimation. Promising results have also been achieved. So, I recommend accept for this paper.